



# Quantification and mitigation of the airborne limb imaging FTIR GLORIA instrument effects and uncertainties

Jörn Ungermann[1], Anne Kleinert[2], Guido Maucher[2], Irene Bartolomé[1], Felix Friedl-Vallon[2], Sören Johansson[2], Lukas Krasauskas[1], and Tom Neubert[1]

[1]Forschungszentrum Jülich GmbH, Jülich
[2]Institute of Meteorology and Climate Research, Karlsruhe Institute of Technology, Karlsruhe, Germany

**Correspondence:** Jörn Ungermann (j.ungermann@fz-juelich.de)

**Abstract.** The Gimballed Limb Observer for Radiance Imaging of the Atmosphere (GLORIA) is an infrared imaging FTS spectrometer with a 2-D infrared detector operated on two high flying research aircrafts. It has flown on eight campaigns and measured along more than $300\,000\,\mathrm{km}$ of flight track.

This paper details our instrument calibration and characterization efforts, which in particular leverage almost ex-
clusively in-flight data. First, we present the framework of our new calibration scheme, which uses information from all three available calibration measurements (two blackbodies and upward pointing "deep space" measurements). Part of this scheme is a new correction algorithm correcting the erratically changing non-linearity of a subset of detector pixels and the identification of remaining bad pixels.

Using this new calibration, we derive a 1-$\sigma$ bound of 1% on the instrumental gain error and a bound of $30\,\mathrm{nW\,cm^{-2}sr^{-1}cm}$
on the instrumental offset error. We show how we can examine the noise and spectral accuracy for all measured atmospheric spectra and derive a spectral accuracy of $5\,\mathrm{ppm}$, on average. All these errors are compliant with the initial instrument requirements.

We also discuss, for the first time, the pointing system of the GLORIA instrument. Combining laboratory calibration efforts with the measurement of astronomical bodies during the flight, we can derive a pointing accuracy of
$0.032°$, which corresponds to one detector pixel.

The paper concludes with a brief study on how these newly characterised instrumental parameters affect temperature and ozone retrievals. We find that, first, the pointing uncertainty and, second, the instrumental gain uncertainty introduce the largest error in the result.

## 1 Introduction

The upper troposphere/lowermost stratosphere is a region of strong chemical contrast and a highly differentiated vertical structure. Its composition is determined by various processes, e.g., convection, lightning, biomass burning,



aircraft exhaust and stratosphere-troposphere exchange (Holton et al., 1995) and it also heavily influences surface climate due to its impact on radiative transport (Forster and Shine, 1997; Riese et al., 2012; Xia et al., 2018).

Historically, satellite limb sounding instruments have served well in observing this atmospheric region (e.g., Hegglin et al., 2009, 2021). A closer and more highly resolved view is feasible by moving the measuring instrument much closer to the region of interest. Air-borne limb sounders allow the high measurement density needed to resolve the fine structures (e.g., Krasauskas et al., 2021).

Principally, limb sounders offer a high vertical resolution but lack in horizontal resolution as the technique does
not allow resolving structures along the lines of sight of the instrument. Besides 2-D cross-sections along the flight path, the Gimballed Limb Observer for Radiance Imaging of the Atmosphere (GLORIA; Riese et al., 2014; Friedl-Vallon et al., 2014) also allows for tomographic reconstruction of limited volumes of interest (e.g., Ungermann et al., 2011; Krisch et al., 2017) by combining measurements in different azimuthal view directions thereby overcoming the conventional limitations of the measurement technique.

The GLORIA instrument has been flown successfully on several measurement campaigns, from TACTS/ESMVal in 2012 (e.g., Rolf et al., 2015), PGS in 2015/2016 (e.g., Woiwode et al., 2018), over StratoClim (e.g., Höpfner et al., 2019) and WISE (e.g., Kunkel et al., 2019) in 2017 to the most recent SouthTRAC campaign in 2019 (Johansson et al.). During 600 flight hours in total, GLORIA collected more than 120 TiB of data providing ≈300 000 km of atmospheric profiles and many volumes resolved in three-dimensions. The quality of the measured infrared radi-
ances was sufficient from the beginning to allow the retrieval of temperature and strongly emitting trace gases from TACTS/ESMVal onward (e.g., Ungermann et al., 2015; Woiwode et al., 2015; Johansson et al., 2018).

Over the course of processing and evaluating data from these campaigns, our understanding of the instrument has constantly improved (Kleinert et al., 2014; Guggenmoser et al., 2015; Kleinert et al., 2018). We have consequently adapted and standardized our in-flight calibration scenario. As the instrument behaves differently aboard the aircraft
than in the laboratory, we focus more and more on using solely in-flight data for characterization efforts. And the continuously growing dataset provides an excellent basis for a profound in-flight instrument characterization.

The structure of this paper is as follows: The first part of this paper starts with a brief instrument description and introduces the current calibration and processing scheme for converting GLORIA measurements to radiance spectra by exploiting all three available calibration sources: the two on-board calibration blackbodies as well as up-
wards pointing "deep space" measurements. This includes a correction for the non-constant non-linearity of several detector pixels as well as a bad pixel identification and some other effects.

Second, we use the in-flight measurements to quantify the major error sources, or at least derive the upper bounds of these errors, and characterize important instrument parameters such as relative and absolute radiance accuracy, radiance precision (noise equivalent spectral radiance; NESR), spectral accuracy, line of sight, and point spread
function.





Third, we briefly discuss the impact of the characterized errors on level 2 products using a simple temperature and ozone retrieval as example. Thus, this study collects all relevant processing information for GLORIA in one place, thus forming a reference for further geophysical interpretation of the data or derivative satellite-borne instruments.

As such, this study collects all relevant processing information for GLORIA in one place, thus being a reference

for further geophysical interpretation of the data or derivative satellite-borne instruments.

## 2  The GLORIA instrument

The GLORIA instrument consists of a spectrometer and a gimbal mount that stabilizes the spectrometer in the airborne environment and allows for pointing the instrument's line of sight (LOS) either at the atmospheric limb, at a 10° upward angle, at nadir, or at one of the two integrated blackbody calibration sources. The spectrometer

combines a classical Michelson interferometer with a two-dimensional detector, allowing for up to 16 384 interferograms (128 × 128) being taken simultaneously. In the typical configuration, we use 128 vertical by 48 horizontal pixels for a total of 6 144 interferograms. The Michelson interferometer can be configured for an optical path difference of up to $\pm 8\,\text{cm}$, which corresponds to a spectral sampling of $0.0625\,\text{cm}^{-1}$. The instrument uses a cryogenic HgCdTe detector array for the detection of infrared radiation in the spectral range between 780 and $1\,400\,\text{cm}^{-1}$.

We focus on this range in our analysis. Outside this range the data quality deteriorates quickly, but is sometimes still useful for retrievals. Thus, we usually show the slightly larger spectral range from 750 to $1450\,\text{cm}^{-1}$. Atmospheric measurements are typically taken in one of three operating modes listed in Tab. 1 with differing trade-offs between temporal/spatial and spectral resolution. A schematic of the instrument is shown in Fig. 1.

The whole spectrometer is cooled with solid carbon dioxide (dry ice) to about $220\,\text{K}$ in order to reduce the

instrument self-emission and to thus enhance the signal-to-noise ratio. A detailed description of the instrument concept was published by Friedl-Vallon et al. (2014).

The processing chain of GLORIA encompasses the usual level 0 to level 2 structure. The raw radiance measurements are digitized with 14 bit A/D converters and stored in a proprietary binary format in combination with time stamps of zero-crossings of a reference laser that allow for a highly accurate determination of the optical path differ-

ence of the image. The level 0 processing step then transforms the measured data (evenly sampled in time) to proper interferograms (evenly sampled in optical path distance). This step includes a non-linearity correction based on laboratory characterization of the used detector array, and a Shannon-Whittaker resampling as presented by Brault (1996). During the resampling step, the spectral axis of each pixel is scaled according to its off-axis angle in order to account for the effect that the actual optical path difference (OPD) depends on the angle between the light observed

at a detector pixel and the optical axis of the instrument. We typically apply a strong Norton-Beer apodization to all our interferograms, unless specified otherwise (Norton and Beer, 1976, 1977). The level 1 processing step then transforms the interferograms into spectra by means of a Fourier transformation and compensation of the instrumental gain and offset using the complex calibration approach presented by Revercomb et al. (1988). These processing



steps have been described in detail by Kleinert et al. (2014) and Guggenmoser et al. (2015). The level 2 processing

step uses the calibrated spectra to determine geo-located temperature, trace gas volume mixing ratios, and ancillary information, e.g., about aerosol load and cloudiness by means of inverse modelling. In practice, an assumed atmospheric state is varied until simulated measurements for that state agree with the actual measurements. Two different level 2 processors are used for evaluating GLORIA data, depending on the recorded spectral resolution. These have been described by Ungermann et al. (2015) for *high spatial resolution* measurements and Johansson et al. (2018)

for *high spectral resolution* measurements (cf. Tab. 1).

## 3   Radiometric calibration concept

Here, we introduce the overall concept of our revised calibration scheme before describing individual parts in detail in the following sections.

The radiometric calibration is the central step of the level 1 processing. It maps the Fourier-transformed interfer-

ograms (raw complex spectra with arbitrary units) to proper physical quantities. This also includes a removal of all signatures of the instrument's self-emissions (Revercomb et al., 1988).

GLORIA is operated in an open compartment below the aircraft and environment temperatures can carry between less than $230\,\mathrm{K}$ and more than $300\,\mathrm{K}$. The self-emission of the interferometer is significant and changes over time, which requires calibration measurements taken in-flight. For this purpose, the instrument is equipped with two

blackbodies that can be actively heated, cooled, or stabilized at a desired temperature (Olschewski et al., 2013). In order to avoid ice contamination and for an optimized power budget, the "cold" blackbody is stabilized at about 0 to $10\,\mathrm{K}$ below ambient temperature, while the "hot" blackbody is heated to 30 to $40\,\mathrm{K}$ above the cold blackbody. A third calibration source is the so-called "deep space". Here, the instrument is pointed upwards at ten degrees elevation angle to measure the dark background of space, which would deliver effectively a direct measurement

of the instrument offset $L_o$, if the carriers were not flying so low that the atmospheric emissions cannot be fully neglected. In Sect. 4.1, we describe our current method for removing the remaining emission lines to approximate a true deep space measurement. In principle, two calibration sources would be sufficient for the radiometric calibration so the use of three sources allows for redundancy and validation.

The deep space measurements have to be recorded in high spectral resolution mode in order to resolve atmospheric

features, while blackbody measurements can be recorded in high spatial resolution mode at 1/10th of the full spectral resolution and are thus much less time consuming to take. Therefore, we take deep space measurements only about once per hour, while measurements of the two blackbodies are recorded about every 15 minutes. In order to enhance the signal-to-noise ratio (SNR), 10 blackbody spectra for each blackbody (requiring in total $\approx 25\,\mathrm{s}$) and 6 deep space spectra (requiring in total $\approx 85\,\mathrm{s}$) are taken.

We follow the approach of (Revercomb et al., 1988) that we can transform our uncalibrated complex spectra with arbitrary units to calibrated complex spectra by a multiplicative gain and an additive offset term. The uncalibrated



spectra are necessarily complex due to the Fourier transform of our asymmetric real interferograms. Even perfectly calibrated spectra are still complex as part of the (asymmetric) measurement noise is mapped into the imaginary plane. With $L \in \mathbb{R}$ being the radiation measured by the instrument, $g \in \mathbb{C}$ the complex gain, and $L_o \in \mathbb{C}$ the offset

(largely caused by self-emissions of the interferometer), the measured signal $S \in \mathbb{C}$ can be described as

$$S = g\left(L + L_o\right). \tag{1}$$

These terms are all functions of both the detector pixel $(u, v)$ and wavenumber $\nu$ of the spectral sample.

Assuming the response of the detector to be linear with respect to the incoming photon flux (see also Sect. 4.2), the gain $g$ and offset $L_o$ may be determined from two uncalibrated measured spectra $S_1 \in \mathbb{C}$ and $S_2 \in \mathbb{C}$ of blackbody

radiation sources with known emission characteristics.

$$g = \frac{S_2 - S_1}{B(T_2) - B(T_1)}, \tag{2}$$

and

$$L_o = \frac{S_1}{g} - B(T_1) = \frac{S_2}{g} - B(T_2), \tag{3}$$

with $B$ being the Planck function and $T$ the temperature of the corresponding blackbody. If a deep space spectrum

$S_d$ is used as one blackbody spectrum, the radiation from this source is (effectively) zero, and Eqs. 2 and 3 simplify to

$$g = \frac{S_1 - S_d}{B(T_1)}. \tag{4}$$

and

$$L_o = \frac{S_d}{g} = \frac{S_1}{g} - B(T_1). \tag{5}$$

As shown by Kleinert et al. (2018), the calibration parameters determined by two blackbodies with a rather small temperature differences are more susceptible to errors in, e.g., blackbody temperature or homogeneity than the combination of one blackbody and a deep space measurement. Since the radiance from the cold blackbody is closer to (but still above) the radiation coming from the atmosphere, it is better suited for the determination of the gain function than the hot blackbody. We therefore determine the gain once every hour from the cold blackbody and the

deep space measurements.

We found that the gain magnitude, which is governed by the sensitivity of the detector array, is very stable during any given flight (see Sect. 5.2). Therefore, one median gain magnitude is determined for each flight. The phase of the complex gain function varies depending on the direction of the moving slide during interferogram acquisition (called "forward" and "backward") but also slowly changes with time because of the thermal variations of the instrument.

Therefore, the phase is linearly interpolated in time between the calibration measurements.





The instrument offset is dominated by its thermal self-emission and changes during the flight, along with the changing temperature of the instrument. When the gain is known, the offset can be calculated from this gain and the spectrum of one calibration source. Since the recording of the blackbody spectra is much faster and is thus taken more frequently, we use the gain and the cold blackbody measurements to determine the instrument offset
every 15 minutes. Between the calibration measurements, the offset is linearly interpolated in time except for the contribution of the outer entrance window (see Sec. 4.3).

The main differences to the calibration method described by Kleinert et al. (2014) are the use of a technique for removing atmospheric emissions from deep space spectra (see Sect. 4.1), an improved non-linearity correction (see Sect. 4.2), and the averaging of gain magnitude over the flight and real part of the offset over forward and backward
sweep directions.

## 4 Methods

This section presents methods developed to account for effects of a real instrument at (comparatively) low flight altitude. For each effect we will give first a short explanation of the physical cause and then detail the method developed to compensate and characterize this effect in the calibration process.

### 4.1 Removal of atmospheric contribution to calibration spectra

For satellite measurements, the instrumental self-emission can be measured by directly looking into true deep space with effectively zero radiance. Due to GLORIA's position below the aircraft, the maximum elevation angle of its optical axis is $\approx 10°$ above the horizon. Taking deep space measurements at a $10°$ angle, combined with a flight altitude below $15\,\mathrm{km}$, there is still a considerable amount of atmosphere in the line of sight of these deep space
measurements. In order to remove the atmospheric radiance emitted by the air within this line of sight, we use atmospheric retrieval techniques to best model the atmospheric contribution to the measured signal and then subtract the forward calculated atmospheric spectrum from the measured one. As a starting point, we calibrate the deep space measurements using gain and offset functions derived from measurements of the two onboard blackbodies. The median spectrum over the whole detector array is used for the representation of the radiance of the central pixel.
This is justified by the observation that the radiance variation over the detector field at this upward pointing elevation angle is rather linear with the elevation angle or pixel row. The median is insensitive to outliers, hence the bad pixels have a negligible impact on the result.

The forward calculation of the spectra requires a priori assumptions on the atmospheric state, i.e., pressure, temperature, and volume mixing ratios (VMRs) of the relevant trace gases emitting in the spectral range covered by the
180 measurements. Pressure, temperature, $H_2O$ and $O_3$ are taken from ECMWF analysis data (e.g., Dee et al., 2011), linearly interpolated to the measurement location and time. The other trace gas profiles are taken from a standard atmosphere (see Sect. E). Forward calculation and retrieval are performed with the radiative transfer model KOPRA


(Karlsruhe Optimized and Precise Radiative transfer Algorithm; Stiller, 2000) and the retrieval software KOPRAFIT (Höpfner et al., 1998).

The goal is to model at best the atmospheric contributions present in the measured deep space spectra in order to unveil the instrument self-emission. The actually retrieved temperature and trace gas profiles are not employed further as this model is tuned towards fitting the measurements optimally and not to derive realistic atmospheric parameters. The atmospheric spectrum is modeled in the spectral range from 750 to $1400\,\mathrm{cm}^{-1}$, allowing for a good calibration quality also slightly outside the specified spectral range starting at $780\,\mathrm{cm}^{-1}$.

The fit is performed iteratively. First, a residual radiance offset is fitted and subtracted from the measured spectrum. In the next step, a broadband fit is performed with temperature and eight gases ($H_2O$, $CO_2$, $O_3$, $N_2O$, $CH_4$, $HNO_3$, CFC-11, and CFC-12) as fit parameters. Altogether, 29 gases are used in the forward calculation. The fit is then refined by fitting single gas profiles in dedicated microwindows in several iterations. The gases and iterations are shown in Table 2 and described in more detail in Appendix A.

The fit results are then used for a forward calculation over the whole spectral range. The processed spectra, from which atmospheric emissions were removed, are called shaved spectra. The shaved and measured spectra are shown in Fig. 2. Besides some remaining atmospheric features, the residual reveals a Planck-like offset and some broadband structures below $850\,\mathrm{cm}^{-1}$, which can be attributed to the germanium entrance window. The remaining atmospheric features are reduced to the order of $10\,\mathrm{nW\,cm}^{-2}\mathrm{sr}^{-1}\mathrm{cm}$.

The forward calculated spectrum is valid for the central row of the detector, because the median over the array has been fitted. Although the variation with elevation angle is rather small for the upward-looking measurements, it is not fully negligible. Therefore, forward radiative transfer calculations are performed for an elevation of +8° and +12°, corresponding to the lowermost and uppermost detector row, respectively. For the rows in between, these spectra are linearly interpolated (see Fig. 2). The residual is slightly different to the one calculated for the corresponding

elevation angle of +10°, but the differences are much smaller than the remaining atmospheric features and therefore negligible.

The subtraction of atmospheric features from the uncalibrated spectra is done for each pixel individually by interpolating the forward calculated spectra to the corresponding row and multiplying this spectrum by the gain function of the corresponding pixel, using the gain that was determined from the two blackbody measurements.

Several small sections of the spectrum are then linearly interpolated from the neighboring spectral samples, either because of still remaining atmospheric features (namely Q-branches of $CO_2$, $HNO_3$, and $CH_4$) or because of spikes in the spectrum at distinct known frequencies due to electrical noise.

The uncertainty of the instrument offset determination is estimated from the difference between the measurement after subtraction of broadband offset and the forward calculation. It is estimated to $20\,\mathrm{nW\,cm}^{-2}\mathrm{sr}^{-1}\mathrm{cm}$ (2 sigma).

Details of this uncertainty estimation are given in Appendix A.



## 4.2 Detector non-linearity correction

The detector is subject to non-linearity, which causes a change in sensitivity depending on the overall photon load. In a first order approximation, this effect causes a scaling of derived spectra, which depends on the photon load of the scene (see App. B for more details).

As described by Kleinert et al. (2014), the non-linearity of the detector has been characterized by carefully performing dedicated measurements of a constant source while varying the integration time on ground. From these measurements, one correction curve for all pixels was determined. This curve works well for most of the pixels, but a considerable number of pixels show spontaneous changes in their non-linear behavior which makes the derived correction unsuitable for these pixels. We attribute this to the different thermal expansion coefficients between the

detector material and the silicon substrate, causing make-and-break contacts Perez et al. (2005). In earlier data versions, these pixels had simply been filtered out, but rigorous filtering leads to a considerable decrease in the number of usable pixels. Therefore, we developed a method to determine a correct gain for these pixels as well. Extending the work of Guggenmoser et al. (2015), we constructed an algorithm to exploit the smoothness of the instrumental offset $L_o$ to correct the faulty pixels.

As the instrument is focused on infinity, the instrumental offset must be a spatially smooth function, as any features in the image from objects residing within the instrument are effectively folded with a Gaussian with a very large support. Thus, we assume that any spatial discontinuities in $L_o$ are caused by the uncorrected non-linearity of the involved pixels. The assumption on spatial smoothness was exploited by Guggenmoser et al. (2015) to improve upon the offset without correcting also the gain.

The non-linearity causes, in a first order approximation, a scaling of the affected spectra. The atmospheric and deep space measurements are reasonably close in photon load and are thus behave in a very similar fashion. In contrast, blackbody measurements have a much stronger signal and scale differently for the problematic pixels. Thus, we use a non-linear fit to estimate pixel-wise scalar non-linearity scaling factors for the blackbody measurements (see Appendix B for details). We derive these non-linearity scaling factors for adjusting the blackbody spectra such

that the offset $L_o$ is free from discontinuities over the whole spectral range (see (Eq. 4)). A set of such derived non-linearity scaling factors is depicted in Fig. 3. The irregular, clustered distribution of bad pixels is clearly visible. The uncertainty in determining theses factors is slightly larger in the circular region where the instrumental offset is close to zero for most of the spectral range. The uncertainty in determining the factors is in the order of a third to half a percent. Comparing the values derived from the forward and backward sweep direction suggests an uncertainty of

≈0.5% on average. Figure 4 shows the histogram of derived correction factors. The factors cluster around the one value, with a Laplacian-like distribution in the center. However, there is a significant number of pixels with a strong non-linearity in the 5% to 10% region.

   The non-linearity correction factors are derived for each calibration sequence containing a deep space measurement, but are applied also to blackbody measurements in between by linear interpolation in time. As the non-linear



behavior is known to change between flights, it may also change within a flight. We thus analyse a subset of atmospheric spectra for calibration artifacts. Excluding clouds, atmospheric spectra are typically spatially smooth as well.

Using a similar method as described above, we now determine a scaling factor for the gain function. With a perfect correction or instrument, a value of 1 is expected for all pixels. Differences from one can be interpreted as remaining

255 error in gain after non-linearity correction. Analysing a set of 158 atmospheric measurements uniformly distributed over flight 16 of the WISE campaign gives Fig. 5. The determined scaling errors are much lower than the correction factors applied to the raw blackbody spectra. Only the lowermost rows show large errors; these interferograms taken at lower altitudes already exhibit a mean value sufficiently different from those of deep space measurements to be subject to a different point on the non-linearity curve compared to deep space spectra.

The resulting row-averaged scaling errors are within $\pm 1\%$, whereby some of the largest differences are due to individual pixels of high variability and the rows where (coloured) noise is generally quite high. The standard deviation of the remaining error in gain averaged over rows can be thus computed to be $\approx 0.2\%$.

### 4.3 Outer window emission correction

The entire instrument, including all windows, is calibrated in flight using deep space and blackbody measurements.

Thus, emission and absorption by windows should be removed by the calibration process. But if temperatures change rapidly compared to the calibration frequency of about $15\,\mathrm{min}$ the fundamental assumption of a constant instrument offset $L_o$ during the acquisition of calibration spectra does not hold anymore. The only critical component in this context is the entrance window of the spectrometer, because only this component changes its temperature rapidly in respect to the calibration frequency of once every 15 minutes. Especially during ascent, the temperature changes

at a rate of up to $2\,\mathrm{K}$ per minute, while temperature changes of all other components are typically below $200\,\mathrm{mK}$ per minute. The instrumental gain is computed from calibration sequences with a stable instrument and window temperature and averaged over the flight such that only the instrumental offset is affected by this problem.

The spectral emission signature of the germanium window is deduced from in-flight measurements at the beginning of the flight, where the window temperature changes rapidly, while the temperature of the other instrument

components stays rather constant. We attribute the difference of the measured instrument offset between the first two calibration sequences to the change in window temperature and calculate the emissivity of the outer germanium window as

$$\epsilon = \frac{L_{o1} - L_{o2}}{B(T_1) - B(T_2)}, \tag{6}$$

with $L_{o1}$ and $L_{o2}$ being the calibrated instrument offset calculated from the first and second calibration sequence and

280 $T_1$ and $T_2$ being the temperature of the outer window at the measurement times of the first and second calibration sequence.





We did this for several flights, discarding outliers and averaging over the rest. The resulting spectral emissivity of the germanium window is shown in Fig. 6. The impact of the germanium window emission is most readily noticeable in the spectral range around $830\,\mathrm{cm}^{-1}$, where the atmospheric signal is very low, but affects also lower
wavenumbers. The window temperature changes too fast to be captured by the regular calibration measurements after take-off, when the window rapidly cools with dropping environmental temperatures, during and after dive maneuvers, and in situations where the window is exposed to direct sun light. We also found the window temperature to fluctuate by $\approx 0.5\,\mathrm{K}$ during tomographic measurement patterns, where GLORIA quickly points towards different azimuth angles and the window is thus subject to different air flow patterns.

In order to account for these rapid temperature changes of the entrance window, we developed the following approach:

Given two calibration measurements at times $t_0$ and $t_1$, we compute the instrumental offset $L_o$ at time $t$ with $t_0 < t < t_1$ by linear interpolation:

$$L_o(t) = L_{o,t_0}\frac{t_1 - t}{t_1 - t_0} + L_{o,t_1}\frac{t - t_0}{t_1 - t_0} \tag{7}$$

We now add a correction term for the window emission to retain the measured instrumental offset at times $t_0$ and $t_1$ while compensating for the changes in window emission due to the measured window temperature $T_{\mathrm{win}}(t)$ in between. The improved instrumental offset $L_o^*$ is thus computed as

$$L_o^*(t) = \left(L_{o,t_0} + B(T_{\mathrm{win}}(t_0))\frac{\epsilon}{1-\epsilon}\right)\frac{t_1 - t}{t_1 - t_0} + \left(L_{o,t_1} + B(T_{\mathrm{win}}(t_1))\frac{\epsilon}{1-\epsilon}\right)\frac{t - t_0}{t_1 - t_0} - B(T_{\mathrm{win}}(t))\frac{\epsilon}{1-\epsilon} \tag{8}$$

The window emission needs to be enhanced by the factor of $(1-\epsilon)^{-1}$ as the gain function already takes into account
the absorption characteristics of the outermost window. Here, the emission takes place within the instrument and thus, the emission of the outermost window is not attenuated. Please note, that the instrumental offset is subtracted from the measured spectra.

As an example, the effect of the correction is shown for two wavelengths in Fig. 7. Due to the germanium emission characteristics, radiances at $830\,\mathrm{cm}^{-1}$ are strongly affected by the window emission feature, while radiances at
305 $950\,\mathrm{cm}^{-1}$ are mostly unaffected. Figure 7a shows the radiances at 830 and $950\,\mathrm{cm}^{-1}$ for the uncorrected calibration. In situations of strong temperature fluctuations (i.e., after 13:20) the radiances at $830\,\mathrm{cm}^{-1}$ behave very differently from the signal at $950\,\mathrm{cm}^{-1}$. Figures 7c and 7d show the same data when applying the discussed window correction. The radiances at 830 and $950\,\mathrm{cm}^{-1}$ are now much more consistent.

The situation depicted in Fig. 7 shows an untypically large variation in outer window temperature. In this worst
case, the amount of correction applied has a standard deviation of $7\,\mathrm{nW\,cm}^{-2}\mathrm{sr}^{-1}\mathrm{cm}$ at $830\,\mathrm{cm}^{-1}$ and $0.5\,\mathrm{nW\,cm}^{-2}\mathrm{sr}^{-1}\mathrm{cm}$ at $950\,\mathrm{cm}^{-1}$. We assume that only $\approx 90\%$ of the effect can be corrected in this fashion (due to uncertainties in both the window emissivity estimate and window temperature measurements) and that thus a systematic error of at most $1\,\mathrm{nW\,cm}^{-2}\mathrm{sr}^{-1}\mathrm{cm}$ may remain after correction at wavenumbers below $900\,\mathrm{cm}^{-1}$ and one tenth of that above.



## 4.4 Parasitic image correction

As analysed in detail for GLORIA by Sha (2013), reflections of incoming light at the surfaces of the beam splitter cause positive and negative parasitic images because the surfaces of beam splitter and compensator plate are wedged. Typically, the beam splitter is mounted such that these images lie in the horizontal plane and are thus invisible in the horizontally averaged radiance data. During the WISE campaign (and the immediately preceding StratoClim campaign), the beam splitter was turned by 90° and the parasitic images were located on the vertical axis and thus

caused noticeable distortions in averaged data. The magnitude of these parasitic images is in the order of a few percent of the original signal, which introduces significant errors in the vicinity of strong gradients in radiation, i.e., over cloud tops. Therefore we have developed a correction method, which is also applicable if the parasitic images lie in the horizontal plane and the scene is not homogeneous.

The effect is most readily visible in the moon measurements taken for pointing analysis. Figure 8 shows an

exemplary image used to characterize the effect. For simplicity's sake, we assume that the parasitic images can be simulated by a simple convolution of the unperturbed image with a vector with just three non-zero entries, which sum up to one, whereby the center value represents the "correct" image and the outer ones the negative and positive parasitic images, respectively. Under these assumptions, the effect can be corrected nearly perfectly using a convolution of the incorrect image with an inverted vector. At the borders, we simply extrapolate the uppermost and

lowermost row, respectively, for the convolution.

The correction vector contains the position (in whole pixels) and the magnitude of the upper and lower parasitic image, respectively. These four free parameters were determined from six independent moon measurements taken during three separate flights and located at various locations on the detector. The parameters were varied manually until we got a satisfactory correction for all six measurements. We found a common shift of the parasitic images of

16 pixels and a factor of 1.8±0.2% for the negative, upper image and -2.5±0.2% for the positive, lower image. The errors are estimated from the range of visually acceptable values.

Figure 8b shows the corrected image. One sees small remaining artifacts, which could not be fully removed by tuning the four parameters; we believe this to be caused by our overly simple and discrete model of the effect.

To properly quantify the remaining effect after correction, we inspected spectra averaged over the three regions

indicated in Fig. 8. The three spectra represent (A) the background of the atmosphere outside of the region affected by the parasitic image of the moon, (B) inside this region and (C) the corrected version. Figure 9a shows the three spectra. Spectrum B is generally decreased outside the $1\,000\,\mathrm{cm^{-1}}$ region, where the radiance field is vertically homogeneous due to strong ozone emissions. The difference plot of the affected spectrum (B-A) in Fig. 9b shows a discrepancy of up to $\approx 100\,\mathrm{nW\,cm^{-2}sr^{-1}cm}$. The corrected spectrum (C-A) does not exhibit obvious defects above

the NESR level.

The effect is worst at cloud tops, where the radiance can quickly drop from $>5\,000\,\mathrm{nW\,cm^{-2}sr^{-1}cm}$ down to $<500\,\mathrm{nW\,cm^{-2}sr^{-1}cm}$ over a couple of pixels. An underestimation of up to $100\,\mathrm{nW\,cm^{-2}sr^{-1}cm}$ above cloud top



was observed in a couple of profiles. With the assumed uncertainty, the effect is reduced by an order of magnitude to $\approx 10\,\mathrm{nW\,cm^{-2}sr^{-1}cm}$ in the worst case (close to cloud tops), but typically it remains below the noise level.

## 4.5 Bad pixel identification

The behavior of individual pixels of the detector of GLORIA changes significantly from flight to flight to the extent that we need to determine a list of bad pixels for each flight individually. These pixels are then excluded from horizontal averaging when preparing the final level 1 products. The classification between "good" and "bad" is always somewhat arbitrary, and there is a considerable number of pixels which do not belong clearly to either
category. The goal of our bad pixel identification is to identify and discard only the worst pixels that would affect the level 1 product when included. We define good pixels as pixels that agree with the median value of their row. To exclude effects of an inhomogeneous scenery on the one hand and use measurements closely resembling regular atmospheric measurements on the other hand, we decided to analyse the deep space measurements, which are cloud free and available in sufficient quantity for all flights.
For each pixel and deep space measurement, we compute the root mean square error (RMS) between its value and the median of the row over all spectral samples. Analysing each flight individually, we can examine the histograms of the so computed RMS, which always show a very similar structure (Fig. 10): a Gaussian-like peak trailing slowly off to higher values. This is composed of the behavior of "standard" pixels with a Gaussian noise distribution on the one hand, and the influence of other pixels with increased noise or other erratic behavior on the other hand. General
noise level and spread varies from flight to flight and from campaign to campaign due to the configuration, employed electronics, and detectors. We assume that the left side of the distribution in Fig. 10 resembles closely the Gaussian distribution of "good" pixels and fit a Gaussian function with unknown mean, standard deviation, and scaling to it. About 4 percent of pixels are beyond the limits of the plot. The Gaussian curve fits reasonably to the left hand side of the distribution and the peak. We then define those pixels as bad, for which the median of the difference computed
over all deep space measurements is larger than the mean plus 9 times the standard deviation. We use the median here, as defects in the calibration offset of a single calibration sequence could otherwise cause a large number of pixels to be discarded. Figure 11 shows this median (panel a) as well as the masks derived by different thresholds. The 9-sigma threshold is by design very inclusive and the probability to exclude "good" pixels is negligible, as we are not interested in discarding a pixel solely for displaying a slightly increased amount of noisiness. Still, we find
that $\approx 10$ percent of the detector pixels are excluded by this criterion on average. About half of these excluded pixels are obviously defect and lie mostly on the outermost columns, where the read-out electronics has known issues. The other half shows a variety of behaviors; for example, some pixels exhibit a telegraph noise pattern, switching their mean value rapidly between different levels, other change their non-linearity during the flight and have thus an offset during longer periods of time.
Figure 11c shows the finally chosen mask for an example flight. In Fig. 11a, a mask with a stricter 3-sigma threshold is shown for comparison. This mask shows large clusters of masked pixels in the top left corner, which





are likely flagged due to border effects in the smoothing of calibration offset. For reference also a more relaxed 12-sigma threshold is depicted in Fig. 11d. To further motivate the chosen threshold, we also examined both the noise of the level 1 data as well as the quality of level 2 results (trace gases and temperature). All depicted masks perform very well with respect to level 2 results, whereas applying no mask causes artificial horizontal structures in level 2 data. Estimating the average noise of spectral samples of cloud-free pixels gives us Table 3. Employing no filtering increases the average noise value significantly. All thresholds effectively filter out really bad pixels to the extent that the estimated noise is very similar. Due to the irregular distribution of more noisy pixels on the detector, using a strict threshold can decrease the number of remaining pixels in some rows to only a handful, causing the average noise value to increase again. To make the most of the available measurements we thus decided on the 9-sigma threshold.

## 4.6 Pointing analysis

GLORIA makes use of a highly sophisticated and precise pointing system based on high-precision sensors and a gimbal mount providing agility on all three axes. The pointing system enables two different limb view acquisition modes for high spatial and high spectral resolution, nadir pointing, as well as calibration. Different control modes are connected to these observation scenarios, because the requirements are different. The major features of the different control modes and the pointing system are described in Appendix C.

An additional camera operating in the visible spectral domain is mounted on the interferometer. This camera covers a wide field-of-view (FOV), thereby completely enclosing the FOV of the spectrometer. For most of the interferograms, correlated images of the scene or video sequences are taken. The information provided by this camera can be used for cloud identification and for pointing quality analysis.

Limb sounding an its retrieval is strongly dependent of the acquisition and absolute knowledge of the line of sight. Therefore an on-ground calibration is inevitable in order to determine pointing offsets and to get a good pointing acquisition in flight. For this purpose, we have built a calibration optics system delivering parallel beams with a broadband infrared light source and an off-axis parabolic mirror. Since this optical system is made for several purposes like determining FOV or adjusting the focal length of the spectrometer, there are five sources arranged like the "five" on a die, but in fact for the pointing calibration just the source in the middle is used. The sources can be seen both in the visible and the infrared spectral range. The calibration optics is mounted on a tripod and the beam of the middle light source is adjusted with the help of a theodolite to be horizontal with an accuracy of $\approx 0.05\,\mathrm{mrad}$. This optical system being placed in front of the instrument pointing towards the spectrometer allows for determination of the offset angle between the nominal horizon of the gimbal control and the real horizon (see Fig. 12). This offset value is commanded to the control system, so that the output of measured elevation is referenced to real horizon. The image of the source on the GLORIA detector has a size of 3 to 5 pixels (see Fig. 13). We assume an uncertainty of 1 pixel for determining its center, which corresponds to $\approx 0.03°$.





With the equipment described above, this measurement can be made for any elevation value. For the offset in azimuth this is more difficult. The best azimuth calibration can be performed in the hangar of HALO at DLR. In this home base hangar of HALO, TU Dresden measured the precise coordinates of reference points marked on the hangar walls and also on the floor close the the position of the integrated GLORIA (Scheinert and Barthelmes, 2014). With these marks, it is possible to validate absolute azimuth and elevation with respect to the pointing system of GLORIA.

In the laboratory at KIT there are known positions of some landmarks too, so the azimuth can be determined there as well but not as accurately as in the hangar at DLR. We thus assume a higher uncertainty in azimuth of $\approx 0.1°$.

    Due to the high demands of limb sounding, the need for determination the absolute pointing knowledge in flight arises. The on-ground calibration has to be verified and corrected, because the absolute pointing of the instrument typically changes due to thermal warping between ground and flight conditions. This absolute attitude might also

change from flight to flight during a campaign, for instance due to the forces working on GLORIA during landing of the aircraft. Sometimes, mis-configurations of the instrument or the exchange of the navigation system with the spare have a similar effect.

    In order to perform an in-flight LOS calibration, a suitable astronomic object has to be observable by GLORIA, i.e., close to the horizon and on the right side of HALO. During WISE several dedicated observations of moon-rise

and moon-set were taken, providing an absolute calibration source under flight conditions. Since flight time and path of the aircraft were determined by scientific goals, such measurements were only feasible during three flights of the WISE campaign as secondary objectives. One such measurement is shown in Fig. 14. Refraction of the visible light close to the horizon impacts the apparent position of the moon. Our approach to correct for this is described in Appendix D. Analysing measurements of past campaigns revealed some previous accidental moon measurements,

whereas campaigns after WISE continue to take moon calibrations whenever feasible. The visible camera can easily locate Venus and other planets visible to the naked eye, but the derived LOS from such measurements comes along with the added uncertainty of alignment between visible and IR camera. In the infrared, the planets are much less bright, but we could successfully identify Venus after averaging 48 images providing an alternate target for line-of-sight calibration.

In addition to these direct but sparse pointing measurements, we perform level 2 retrievals to determine the attitude. Here, we use data of two such retrievals: one based on data acquired with high spectral sampling ($0.0625\,\mathrm{cm}^{-1}$; Johansson et al., 2018) and one based on data acquired with intermediate spectral sampling ($0.2\,\mathrm{cm}^{-1}$; see Appendix F for details), which is available for all profiles. For campaigns prior to 2017, for some flights only data in coarse spectral resolution is available ($0.625\,\mathrm{cm}^{-1}$). For these flights a different approach based on the retrieval

described by Ungermann et al. (2015) was used, where instead of temperature an elevation correction value was derived (the $CO_2$ lines used starting from WISE are not sufficiently resolved in the early campaigns). We typically use only a single correction factor for a flight, which is determined after filtering short (less than $3\,\mathrm{km}$ between instrument and cloud top) profiles from the mean of the remaining values. An error estimate is computed from the standard deviation.





All line of sight characterizations for the WISE campaign are aggregated in Fig. 15. The attitude was calibrated using the on ground calibration before and after the campaign. Between WISE flights 2 and 3, the inertial navigation system needed to be exchanged with the spare, causing a change in elevation offset. Between flights 13 and 14, the pointing was readjusted using information from preliminary level 2 and moon data that suggested a systematic offset of 0.17° at the time. Thus, the pre-campaign calibration is applied for flights 1 and 2 and the results of the

post-campaign calibration are applied for the remaining flights, respectively. The values of the different level 2 retrievals agree within the respective error bars. In this particular campaign, the moon calibration seems to indicate a systematically smaller elevation correction, but other campaigns show also higher values. The differences between the various methods and calibrations are consistent within the estimated uncertainties. For the further processing we typically select the most reliable value (depending on number of available profiles and spectral resolution) from the

level 2 result. As remaining error we assume simply an uncertainty of one pixel, i.e., ≈0.032°.

## 5   Performance and Characterization

This section gives an analysis of the quality of our level 1 data from in-flight data and aggregates the (simplified) result in a table to serve as a basis for error estimates of level 2 products such as temperature or trace gas VMRs.

### 5.1   Noise equivalent spectral radiance (NESR)

Friedl-Vallon et al. (2014) estimated the NESR of GLORIA from selected measurements to assure it is within specification. Here, we extend this work to estimate the noise of individual atmospheric measurements for different altitudes and flights. For the level 2 processing we need to determine the NESR associated with spectra averaged over an entire row. We thus focus on the NESR of spectra averaged over the detector rows.

There are several methods to estimate the NESR from measured spectra. In an ideal instrument, the imaginary part

of calibrated spectra contains only measurement noise. In practice, however, it also contains some residual signal due to, e.g., small phase errors (the effect of which is negligible in the real part of the spectrum), asymmetries in the interferogram due to a variable scene (especially in presence of clouds), and artifacts introduced by calibration inaccuracies in the imaginary part of the instrument offset. Instead, we are using two different methods to determine the NESR from the real part of the spectrum.

For the first method, we look for each detector pixel at the radiance variation over seven consecutive measurements of a deep space sequence. We can safely assume that the observed scene stayed constant during this brief time frame. These pixel-based estimates are used to determine the NESR of horizontally averaged values using the bad pixel mask determined in Sect. 4.5. Then, the resulting NESR spectra are averaged vertically to present a single spectrum.

For the second method, we only use a single deep space measurement and look at the horizontal variation from

detector pixel to detector pixel. In particular, the NESR is estimated by computing the horizontal standard deviation (again excluding bad pixels as determined according to Sect. 4.5) for the first measurement of the sequence only.





These values are then divided by the square root of the number of corresponding valid horizontal pixels and averaged as above over the vertical dimension.

Results from both methods plotted against wavenumber are compared in Fig. 16. These noise spectra were computed for deep space measurements processed in the full $0.0625\,\mathrm{cm}^{-1}$ spectral resolution of the high spectral resolution mode as well as the reduced $0.625\,\mathrm{cm}^{-1}$ of the high spatial resolution mode, which we also use for blackbody measurements. The NESR spectra derived from horizontal variation are 10% and 5% higher than those derived from temporal variation. This is partially expected as calibration noise and calibration inaccuracies only contribute in the horizontal variation analysis. We also observe additional structures in the ozone $1000\,\mathrm{cm}^{-1}$ band, which we associate with imperfections in the shaving. Locally enhanced values in the NESR are caused by electrical disturbances. The highly resolved NESR spectra are 3.14 and 3.03 times higher than those derived from low resolution data for the time and horizontal method, respectively. This is reasonably close to the expected factor of $\sqrt{10}$ such that the NESR can be estimated from measurements with either resolution and scaled to any different resolution employed in atmospheric observations.

Employing the second method, we can now produce an NESR estimate for all of our atmospheric measurements allowing us to closely track instrument performance over the whole flight. Figure 17 shows an example of such an analysis for a set of roughly 200 evenly distributed measurements from one flight. Figure 17a is depicting an NESR averaged spectrum; only spectra determined as fully cloud free (having a cloud index of more than 6 according to (Spang et al., 2012)) are included in this average. Figure 17b shows the evolution of noise over time as a pseudocolor plot. The NESR is spectrally averaged in the range from $750\,\mathrm{cm}^{-1}$ to $1450\,\mathrm{cm}^{-1}$. The beginning of the flight before 14:00 shows increased NESR values, which can be attributed to the higher blackbody and outer window temperatures. The higher blackbody temperatures at the beginning of the flight require a shorter integration time for the calibration measurements, which are thus subject to a higher NESR, affecting the calibration quality. The high values in the lower part of the detector array are due to clouds which lead to spatial inhomogeneities and thus to an overestimation of the NESR. The last panel (Fig. 17c) shows the noise of individual rows averaged over cloud free measurements. The lowermost rows are missing since no cloud-free spectra were available for analysis. Some high values due to unidentified small-scale clouds remain in the lowermost rows with data. At higher altitudes one can see that the NESR is not uniform over all rows due to the read-out electronics and the uneven distribution of filtered pixels. Typically, all flights of a campaign exhibit very similar NESR characteristics, but from campaign to campaign, we can observe small variations such as a generally increased NESR value due to changes of the instrument or its operation parameters.

From this and analysis of other flights, we derive a typical representation of the NESR with $5\,\mathrm{nW\,cm}^{-2}\mathrm{sr}^{-1}\mathrm{cm}$ for the wavenumber range between $880\,\mathrm{cm}^{-1}$ and $1300\,\mathrm{cm}^{-1}$ and $8\,\mathrm{nW\,cm}^{-2}\mathrm{sr}^{-1}\mathrm{cm}$ outside this range for a spectral resolution of $0.625\,\mathrm{cm}^{-1}$. The figures scale with $\sqrt{3.125}$ for the spectral resolution of $0.2\,\mathrm{cm}^{-1}$ and $\sqrt{10}$ for the spectral resolution of $0.0625\,\mathrm{cm}^{-1}$ (see Table 4).





## 5.2 Gain accuracy

This section estimates the stability of the gain magnitude from in-flight data. During the calibration, we compute a gain magnitude for each calibration sequence containing a pair of blackbody and deep space measurements, each giving an independent gain estimate (see Sect. 3). These estimates are averaged in a succeeding step to reduce 520 the impact of measurement noise. The variability of these gain magnitudes gives us an uncertainty estimate by computing the standard deviation of the gain magnitude for all flights and all pixels. The median standard deviation over all pixels is shown in Fig. 18 for several flights of the WISE campaign. The resulting accuracy is better than 0.1% and thus well within our target range of 1%. One can see an increased uncertainty towards the edges of the usable wavenumber range, which is caused by the decreased sensitivity of the detector and in consequence a 525 lower SNR. Towards lower wavenumber regions, one can also see a spectral structure which resembles the window emission feature shown in Fig. 6; these are likely caused by small temperature variations of the window during the rather long deep space measurements. Around $1050\,\mathrm{cm}^{-1}$ and $1300\,\mathrm{cm}^{-1}$ an increased uncertainty at the location of strong ozone and methane emissions is observed, which is most probably due to imperfections in the shaving of deep space spectra. While these estimates show the uncertainty of an average single detector pixel, we also use them 530 to describe the accuracy of horizontally averaged data, because some of the underlying errors are strongly correlated (such as the impact of atmospheric window emissions). We pick the highest uncertainty of the flights to have a single value for analysis as a worst-case assumption (max).

In addition to these variations, our gain might be subject to a systematic error common to all measurements. There is a range of potential sources for such a systematic error to be stemming from, for example, inaccurate 535 blackbody temperatures, uncorrected detector non-linearity, or shaving defects. To gain an independent estimate of the absolute accuracy of the gain magnitude, we turn to atmospheric measurements and compare the calibrated spectra with a Planck curve of ambient temperature in a wavenumber range which is located within the optically thick ozone Q-branch from $1050\,\mathrm{cm}^{-1}$ to $1056\,\mathrm{cm}^{-1}$. We selected only the atmospheric profiles for which ECMWF indicated an ozone volume mixing ratio above $300\,\mathrm{nmol\,mol}^{-1}$ to ensure sufficient optical thickness and thus prac- 540 tically no dependence on actual ozone VMRs. Computing the relative difference between calibrated measurements and the Planck curve indicated by ECMWF temperatures for 3 000 WISE profiles shows an average difference of 0.1%±2.0%. The rather high uncertainty shows the sensitivity of the estimate to ambient temperature, where ECMWF in the used resolution of 1°×1° cannot follow the small-scale fluctuations of the atmosphere. A similar analysis for other campaigns gave slightly larger differences. For SouthTRAC data, a difference of -0.9%±2.5% 545 was identified and for PGS, a difference of -1.3%±2.6%. In all cases, we could not reject the null hypothesis that there is no gain bias in GLORIA data. As both offset and gain errors would reflect in this analysis as well as the inherent uncertainty in ECMWF temperatures, it is difficult to quantify exactly how accurate the gain is. Instead, we can give an upper bound of 2%, which fits well to our threshold requirement (Friedl-Vallon et al., 2014). For level 2





error estimates, we assume a general gain magnitude error of 1% , which obviously might imply higher or lower
errors in individual flights.

## 5.3   Offset accuracy

The instrumental gain and offset are determined using a direct measurement of the deep space background. Thus, we
expect only small, correlated errors due to measurement noise and imperfect shaving of atmospheric contributions.

In order to quantify these errors in offset, we analysed special in-flight measurements, where the elevation angle
of the optical axis of GLORIA alternated between -0.38° and -1.00° for 12 consecutive images. We interpolated
each profile to an evenly spaced elevation axis, computed differences between succeeding measurements and finally
averaged the differences. The result is depicted in Fig. 19. Within the overlapping range, we find a difference of
$(-1.7\pm5.3\,\mathrm{nW\,cm^{-2}sr^{-1}cm})$ between the different pitch angles. Although statistically not significant, the higher-
pointing measurements are colder, which would be consistent with warm stray light from below affecting the mea-
surements. The available measurements do not allow for a full quantification of the effect, though. Analysing the
discrepancies in more detail reveals spectrally and spatially correlated structures of up to $6\,\mathrm{nW\,cm^{-2}sr^{-1}cm}$ magni-
tude above $900\,\mathrm{cm^{-1}}$ increasing towards $16\,\mathrm{nW\,cm^{-2}sr^{-1}cm}$ below (Fig. 19b). The differences around $1000\,\mathrm{cm^{-1}}$
and other strong emission features may partially be caused by errors in gain, not offset. The magnitude of the dif-
ference is similar for both the upwards and downwards pointing pixels. This lends weight to the hypothesis that it
is largely caused by an offset error, as the measured radiances are much higher at lower pixels which would lead to
higher absolute differences in case of gain errors. Due to the various smoothing methods employed in generating the
offset calibration data, we also expect both spatial and spectral correlation. We assume here an eyeballed correlation
length of 10 pixels vertically and $50\,\mathrm{cm^{-1}}$ spectrally and average the magnitude to $10\,\mathrm{nW\,cm^{-2}sr^{-1}cm}$. This error
is separate from the systematic uncertainty from shaving.

## 5.4   Spectral accuracy

The spectral axis of our level 1 data depends on the proper association between taken images and optical path
difference. We perform an off-axis correction and characterize the laser wavelength as described by Kleinert et al.
(2014), using the deep space measurements. These are cloud free and taken periodically in high spectral resolution
mode.

While the geometric parameters, which determine the off-axis angle, stay constant during one flight, we found
that the laser wavelength sometimes varies significantly, e.g., because of temperature drifts and resulting laser mode
hops. In order to obtain a better temporal resolution of the evolution of the laser wavelength, we modified our
spectral calibration algorithm to determine only the laser wavelength from otherwise off-axis corrected calibrated
atmospheric spectra. This allows us to do quality checks on calibrated and horizontally averaged level 1 data and
to better quantify our uncertainty. For each atmospheric measurement, all pixels unaffected by clouds are averaged,
and the resulting spectrum is analysed for a spectral shift. Due to the high SNR ratio, this spectrum allows for





a reliable spectral shift determination also from data measured with decreased spectral resolution. This enables a continuous monitoring of the laser wavelength over the flight, as many atmospheric measurements are available only at $0.625\,\text{cm}^{-1}$ or $0.2\,\text{cm}^{-1}$ resolution. The method is based on locating the positions of $CO_2$ emission lines in the $950\,\text{cm}^{-1}$ wavenumber region and comparing these with expected line positions of the HITRAN database (Gordon et al., 2017). Estimating the spectral accuracy of $0.625\,\text{cm}^{-1}$ spectra requires dedicated processing run using a Norton-Beer weak apodization instead of the usually employed Norton-Beer strong apodization (Norton and Beer, 1976, 1977).

Figure 20 shows the resulting spectral shifts for an exemplary flight of the WISE campaign. The most precise results are from spectra of highest spectral resolution. The spectra with $0.2\,\text{cm}$ resolution have a larger spread, whereas the $0.625\,\text{cm}^{-1}$ spectra have large errors of about 5–10 ppm (i.e. a relative error in wavenumber knowledge of less than 0.001%) and are thus only useful for a qualitative analysis (to detect large errors). For this flight, the spectral accuracy is in the order of 1 ppm on average. Using the $0.2\,\text{cm}$ data, the accuracy is still diagnosed to be better than 2 ppm. This is much better than the original target accuracy of 10 ppm. Applying this technique to all flights of the WISE campaign, we estimate the spectral accuracy to be in the order of 2 ppm. The same accuracy is also valid for other campaigns from PGS onward with the exception of a few flights subject to known technical issues. To account for additional variations over the detector and to include outliers, we use an accuracy estimate of 5 ppm (see Table 4). This corresponds to about one tenth of the smallest employed spectral sampling distance of $0.0625\,\text{cm}^{-1}$ at the largest wavenumber of $1\,400\,\text{cm}^{-1}$.

## 5.5 Point spread function

The point spread function (PSF) determines the amount and direction of incoming light being measured by the detector pixels. The theoretical shape for a diffraction limited instrument is the Airy-disk:

$$a(r) = \left(\frac{2J_1(\nu Dr)}{\nu Dr}\right)^2, \qquad\qquad \text{for}\quad r > 0$$

$$a(r) = 1, \qquad\qquad \text{for}\quad r = 0$$

with $J_1$ being the Bessel function of first kind, wave number $\nu$, aperture $D$ and distance to the optical axis $r$.

This function defines the PSF for a pixel of infinitesimal extent and needs to be integrated over the pixel size to determine the actual PSF for the pixel. In the horizontal direction, because of our averaging over full rows, we can assume effectively infinite extent with only a small error, but vertically the detector pixel width needs to be considered.

The optical aperture diameter is $3.6\,\text{cm}$. Figure 21 shows several PSF functions for an aperture of 3.2, 3.6, and $4.0\,\text{cm}$ in the upper panel. The lower panel demonstrates how the PSF changes with wavenumber.

We determine the PSF from extinction values from level 2 retrieval results. In the presence of optically thick cloud tops, artifacts will appear in retrieved extinction values if the simulated PSF is different from the real one.





With a PSF that is too wide, the modeled radiance above cloud top is overestimated and, accordingly, unnaturally small extinction values are generated in order to properly simulate the measured radiances. With a PSF that is too narrow, there is no expected sharp step in the extinction profile at the cloud top, but instead a more gradual increase in the transition region. This is exemplified in the extinction profile in Fig. 22 retrieved with different aperture sizes based on the retrieval approach discussed by Ungermann et al. (2020). For a PSF with an assumed aperture smaller than $3.6\,\text{cm}$, extinction drops even below the background values found at higher altitude values above the cloud top. For this specific profile only from an aperture of $3.6\,\text{cm}$ upwards, a sensible extinction profile can be derived. While the range of admissible apertures varies from profile to profile, all derived extinction profiles were plausible for an aperture of $3.6\,\text{cm}$. The same experiment was performed at a different atmospheric window at $1214\,\text{cm}^{-1}$. This shows similar results for the corresponding PSF, so the retrieval results are consistent with the expected value from instrument design. From the different characterization methods and the variability of the results, we assume an uncertainty of 10% in the width of the PSF.

## 6 Analysis of impact on level 2 data

The comprehensive characterization of leading level-1 errors from in-flight data is a solid basis to revisit our level 2 results and estimate the impact of these errors on our derived quantities.

The retrieval of geolocated physical quantities from limb measurements is a ill-posed non-linear problem. Using a forward model simulating the radiative transfer, one adjusts geophysical quantities such as temperature or trace gas volume mixing ratios until the simulated radiances agree with the measured ones within expectation (e.g., Rodgers, 2000). For efficiency, we typically apply a linearized Gaussian error analysis after deriving an atmospheric profile (e.g., Ungermann et al., 2015; Johansson et al., 2020). Here we use a simpler approach to allow also for the quantification of all errors estimated in this paper including those for which the forward model does not offer the necessary derivatives.

For brevity, we examine only two quantities. We selected temperature as it is the fundamental quantity necessary for all further retrievals and ozone as one of the most commonly retrieved trace gases. The corresponding retrieval setups are described in Appendix F. We assume here that the given errors, with exception of the NESR, are systematic and affect all measurements of one flight in the same way. We picked flight 10 (7th October 2017) of the WISE campaign for this exercise as it offers many profiles reaching deep into the troposphere.

- To examine the effect of NESR, we applied independent Gaussian noise of $14.4\,\text{nW}\,\text{cm}^{-2}\text{sr}^{-1}\text{cm}$ to all samples (employed spectral resolution of the level 1 data is $0.2\,\text{cm}^{-1}$).

- The effect of the deep space shaving error is estimated by adding $20\,\text{nW}\,\text{cm}^{-2}\text{sr}^{-1}\text{cm}$ to all measurements.

- The effect of the detector non-linearity is estimated by generating a Gaussian distributed gain error with mean $0\,\%$ and standard deviation $0.2\,\%$ for each pixel and modifying the radiances accordingly.





- The offset error has a spatial structure and to best capture this, we decided to simply apply the difference from Fig. 19a to each measurement of the flight.

- For examining the total gain error we modified the gain in a dedicated level 1 run with a consistent 1% error (-1% gives similar results, but of opposite sign).

- We modified the LOS by 0.032° for the flight (-0.032° gives similar results, but of opposite sign).

- For estimating the effect of PSF width uncertainty, we examined a retrieval with a PSF of reduced width by 10% (increased width behaves quantitatively similar).

- Last, for examining the effect of a spectral shift, we applied a shift of 5 ppm in a dedicated level 1 run by modifying the employed laser wavelength by 5 ppm (-5 ppm behaves quantitatively similar).

We then performed temperature retrievals based on the $CO_2$ laser lines around $950\,\mathrm{cm}^{-1}$ of the whole flight for 191 uniformly spaced profiles (our standard selection for performing test retrievals for this flight) and computed the mean and standard deviation of the difference to an unperturbed reference run. The results are collected in Table 5. Several of these errors have a noticeable structure in altitude, which we neglect (average away) here to simplify the discussion.

The instrument pointing has the largest impact with a mean error of $\approx 1\,\mathrm{K}$. The error close to flight level is thereby smaller but increases towards lower altitudes. Due to the typically employed LOS retrieval based on ECMWF temperatures, differences to ECMWF are typically smaller than this (Johansson et al., 2018), but $1\,\mathrm{K}$ seems a reasonable estimate for the absolute accuracy of the temperature product. The next leading error sources are related to the shaving-induced offset error and gain uncertainty. Both are in the order of $0.25\,\mathrm{K}$. The other errors are an order of 665    magnitude smaller.

The exemplary trace gas retrieval for $O_3$ derives volume mixing ratios from emissions of the $1\,000\,\mathrm{cm}^{-1}$ ozone band. The results are here similar to those of the temperature retrieval. First, larger differences in temperature induce an error itself and, second, the error source affects the ozone retrieval as well. Due to the inclusion of the strong emissions at $980\,\mathrm{cm}^{-1}$, the gain error affects this retrieval especially strongly. All errors are smaller than $5\,\%$, which 670    is often a reasonable accuracy estimate simply due to given uncertainties in spectral line strength and other modelling assumptions.

These error estimates are based on the new, comprehensive data basis of in-flight characterization and validation of our level-1 data quality assumptions. They are in broad agreement with prior estimates by Ungermann et al. (2015) and Johansson et al. (2018), which could rely on ground-based characterization only and hence detail and 675    corroborate these previous studies.





## 7   Conclusions

The GLORIA instrument has been operated for more than ten years now. In this time, we have learned about the new challenges posed by 2-D FTIR imaging observations in general. This enabled us to make full use of all calibration data, to develop methods to statistically use atmospheric observations for in-flight validation and to tackle a number

of known instrument artifacts, several of which are specific to the 2-D concept. This mitigates some of the leading error-sources in previous data versions such as detector non-linearity and pointing.

In this study we have collected the matured state of our calibration and level 1 processing, which forms the basis for further level 2 processing. We have exploited all three available calibration sources (the two blackbodies and the deep space measurements) to correct for the non-linearity of our detector, which can change erratically

between different cold-runs of the detector. This fully solves a problem only partially addressed by previous efforts (Guggenmoser et al., 2015). The new correction allows us to use more of the available pixels thus reducing the NESR of row-averaged spectra.

The described algorithm is able to track the noise-level of measurements during the flight to measure the impact of, e.g., warm blackbodies with short integration times on the calibrated data and identify technical issues. The

690 computed noise level of $5\,\mathrm{nW\,cm^{-2}sr^{-1}cm}$ for $0.625\,\mathrm{cm^{-1}}$ is within our instrument specification.

We determine the line of sight calibration by both immediate measurements of celestial bodies and a level 2 product employing ECMWF temperatures. This allows to put strong bounds on its accuracy of $0.032°$, which corresponds to one vertical pixel.

Further, we leveraged different in-flight atmospheric measurements to get direct bounds on the accuracy of in-

695 strument gain and offset. The gain is accurate within 1%, whereas the offset is subject to a potential bias of up to $30\,\mathrm{nW\,cm^{-2}sr^{-1}cm}$.

These aggregated errors allow us to propagate the error assumptions to level 2 products and identify the leading errors. The analysis show-cases that the largest contributor of uncertainty is the line of sight calibration, which is already reduced to the order of $\pm1$ pixel and thus at the limit of what we can achieve with the given instrument; it

also matches with the initial design specification (Friedl-Vallon et al., 2006). This uncertainty is closely followed by the uncertainty in total gain and offset. Still, both errors are within our initial instrument target thresholds (Friedl-Vallon et al., 2014) and their impact on level 2 data is comparable to other level 2 error sources such as spectroscopic uncertainties. This will allow us to scientifically exploit the data as intended to gain new insights into the chemistry and dynamics of the UTLS region of the atmosphere. But it also demonstrates the capabilities of this instrument

concept in general and the options an imaging detector offers to increase the data quality.

*Data availability.* GLORIA level 1 data are available on request. GLORIA level 2 data are accessible via the (HALO database). The NOAA data are available from the NOAA websites (Dlugokencky and Tans, 2020; Dlugokencky, 2020; NOAA).





**Appendix A: Details on the procedure of removing atmospheric signatures from the measured deep space
spectra**

This section describes in detail the different steps for the fit of the atmospheric signatures in the measured deep space
spectra and the reconstruction of the forward calculated and gives an estimate for the uncertainty of this method.

The spectra which were calibrated based on the blackbody-blackbody measurements still show a significant broad-
band offset which cannot be attributed to the atmosphere. Although the origin of this offset is not completely ex-
plained, it has turned out that it can be well described by a Planck function. This Planck-shaped offset is determined
and subtracted in a first step. For this purpose, the offset is determined at 5 spectral microwindows as listed in Ta-
ble 2, and a Planck function is fitted to these 5 points with temperature and emissivity as fit parameters. In order
to obtain more stable fit results, a scale is also fitted to the data in order to compensate non-perfect VMRs in the a
priori profiles. Furthermore, a spectral shift is fitted in each microwindow, from which a mean shift over the whole
spectral range from 750 to $1\,400\,\mathrm{cm}^{-1}$ is deduced. This shift is used for all further retrievals.

Figure A1 shows an exemplary median deep space spectrum, calibrated with the two blackbody measurements,
the offset values determined in the 5 microwindows and the fitted Planck function. All further plots in this section
also refer to the same measurement.

In the next step, a broadband fit is performed in order to describe the measured spectrum. In total, 29 gases are
considered in the forward calculation. The result of this fit is shown in Fig. A2. In general, the spectrum is well
represented by the fit, and the residual is mostly below $20\,\mathrm{nW\,cm}^{-2}\mathrm{sr}^{-1}\mathrm{cm}$. There are, however, some remaining
atmospheric features of, e.g., $HNO_3$ around 850 to $900\,\mathrm{cm}^{-1}$, $CO_2$ below 800 and around $950\,\mathrm{cm}^{-1}$, $O_3$ around
$1\,050\,\mathrm{cm}^{-1}$, and $N_2O$ and $CH_4$ at higher wavenumbers. In order to reduce these residuals, several species are fitted
again in dedicated microwindows, while the fit results <of all species from the broadband fit are used as new a priori
profiles. For some species, it has turned out that dependencies of interfering species have to be considered. For ex-
730 ample, a simultaneous fit of CFC-11, $HNO_3$ and CFC-12 in the spectral window from 830 to $920\,\mathrm{cm}^{-1}$ did not give
satisfactory results, while a fit of CFC-11 and CFC-12 in a first step, followed by a fit of $HNO_3$ with the fit results
for CFC-11 and CFC-12 as new a priori profiles, considerably reduced the residual. For some species (namely $O_3$,
$N_2O$ and $CH_4$) it was not possible to represent the whole spectral range with one profile. This can be explained with
inconsistencies in spectroscopic data (e.g., Glatthor et al., 2018) or different temperature dependencies. Therefore,
different profiles were fitted for different spectral ranges. The different species and microwindows for these itera-
tions are also given in Table 2. In these iterations, also an offset is fitted for each MW to improve the fit result. This
offset is then discarded.

Figure A3 shows the improvements of the residuals after the individual fits in an exemplary MW.

The final step for the reconstruction of the simulated spectrum is a forward calculation using the fit results of the
740 previous steps. For $O_3$, $N_2O$ and $CH_4$, different profiles are used in different spectral ranges. The $O_3$ profile named
(v2) in Table 2 is used for the spectral range 850–$1065\,\mathrm{cm}^{-1}$, otherwise (v1) is used. (v0) of $N_2O$ is used in the


range 1065–1205 $\mathrm{cm}^{-1}$, (v1) of $N_2O$ and $CH_4$ is used from 1205 to 1285 $\mathrm{cm}^{-1}$, and above 1285 $\mathrm{cm}^{-1}$ (v2) of $N_2O$ and $CH_4$ is used.

For an estimation of the uncertainty of the instrument offset determination, the residuals between the measurement after subtraction of the Planck function and the forward calculation are considered. These residuals are smoothed by a 10 point moving average, because the offset is also calculated from smoothed spectra. In order to provide an overview over all flights and to give a representative uncertainty, the residuals of all deep space measurements of one flight are taken, and the 1-sigma standard deviation is calculated for each spectral point. This gives one uncertainty spectrum for each flight. These data are shown in Fig. A4. In general, the data are quite similar for all campaigns and flights. Only the very early data of the TACTS campaign shows rather large deviations around 830 $\mathrm{cm}^{-1}$. These are attributed to the emission of the germanium entrance window, which is not fully compensated by the calibration using the two blackbodies, because the window correction (see Sect. 4.3) could not be applied to these data, due to a malfunction of the temperature sensor. Therefore, this difference is rather seen as an error in calibration using the two blackbodies and is not attributed to the uncertainty of the instrument offset determination. From Fig. A4, a 1-sigma uncertainty of 10 $\mathrm{nW\,cm^{-2}sr^{-1}cm}$ is derived for the whole spectral range. For the estimation of the systematic uncertainty due to the instrument offset determination, we take the 2-sigma uncertainty, i.e., 20 $\mathrm{nW\,cm^{-2}sr^{-1}cm}$.

## Appendix B:  Pixel-wise non-linearity factor determination

This section describes the computation of pixel specific non-linearity correction factors in detail.

The voltage $U \in \mathbb{R}$ measured by a detector pixel is a monotonous function $f : \mathbb{R} \mapsto \mathbb{R}$ of incoming radiation $P$. Most of the non-linearity has been characterized and is corrected for in an early processing step in the level 0 processing (Kleinert et al., 2014), such that we have here mostly linearly behaving pixels with small variations on top. With $x \in \mathbb{R}$ being the sled position, the incoming radiation in an interferogram varies around the mean $P_0 \in \mathbb{R}$ such that one can linearize $f$ at $P_0$:

$$U(x) = f(P(x)) \approx f(P_0) + f'(P_0)(P(x) - P_0). \tag{B1}$$

For the Fourier-transformation $S$ of the interferogram $U$ follows:

$$S = \mathcal{F}(U) \approx \mathcal{F}(c) + f'(P_0)\mathcal{F}(P), \tag{B2}$$

with $c$ being a constant function influencing only the zeroth frequency. Neglecting higher-order effects, this means, that the uncalibrated raw spectrum is scaled with the slope of the tangent of $f$ at the mean level $P_0$.

As the uncorrected non-linearity only changes slowly with $P_0$, this implies that measurements with similar $P_0$ at the same exposure time will be subject to an effectively identical scaling factor. This applies to optically thin atmospheric and deep space measurements, which are taken at the same exposure time. Problematic are the black-body measurements which require a much shorter exposure time and are subject to a very different photon load or atmospheric measurements affected by strong cloud emissions.





In the final calibration step, we deduce the instrumental gain $g$ from one deep space and one blackbody measurement (see Sect. 3). To derive a correct gain, the different slope induced by the different photon load and exposure time of the blackbody measurements needs to be corrected (our data analysis shows that the deep space measurement is effectively similar in behavior to the atmospheric cloud-free measurements), such that one only needs to determine one linear correction factor $\alpha$ for each detector pixel for a given blackbody measurement $S_{\mathrm{bb}}$, as there is no spectral variation (under the assumption of no effects of higher order). With this linear scaling factor, a corrected gain $g^*$ can be computed as:

$$g^*(u,v,\nu) = \frac{\alpha(u,v)S_{\mathrm{bb}}(u,v,\nu) - S_{\mathrm{ds}}(u,v,\nu)}{B(T_{\mathrm{bb}})(\nu)}. \tag{B3}$$

Here, $S_{\mathrm{bb}}$ and $S_{\mathrm{ds}}$ refer to the raw spectra of shaved deep space and cold blackbody measurement. $u$ and $v$ are the integer coordinates of the detector pixel and $\nu$ relates to a spectral sample.

To estimate the effect of the correction on the calibrated spectrum, one may express the corrected spectrum in terms of the uncorrected one:

$$L = \frac{S}{g^*} - L_o^* = \left(\frac{S}{g} - L_o\right)\frac{S_{\mathrm{bb}} - S_{\mathrm{ds}}}{\alpha S_{\mathrm{bb}} - S_{\mathrm{ds}}} \tag{B4}$$

As the signal $S_{\mathrm{bb}}$ is much stronger than $S_{\mathrm{ds}}$, the non-linearity is effectively causing a scaling of the calibrated spectrum. For our typical blackbody and deep space spectra, an error of 5% in non-linearity causes an error in gain between 4% and 5.2%, varying within this range both spatially and spectrally. These errors scale nearly linearly in the value range observed for our detector. The error assumes its largest values in the spectral regions, where the atmospheric window (see Sect. 4.3) has the strongest emission features and in the centre of the detector, where instrument emission is smallest.

To identify the individual non-linearity, we exploit the fact, that the instrumental offset $L_o$ must be spatially smooth. As such, we define for each detector pixel $(u,v)$ a cost function $J_{u,v}$

$$J_{u,v}(x) = \sum_{\nu} \left( \frac{\mathrm{smooth}\left(\Re\left(\frac{S_{\mathrm{uds}}(\nu)}{g(\nu)}\right)\right)(u,v)}{\Re\left(\frac{S_{\mathrm{uds}}(u,v,\nu)}{g(u,v,\nu)}\right)} - \left|\frac{xS_{\mathrm{bb}}(u,v,\nu) - S_{\mathrm{ds}}(u,v,\nu)}{S_{\mathrm{bb}}(u,v,\nu) - S_{\mathrm{ds}}(u,v,\nu)}\right| \right)^2 + \lambda\|1-x\|_2^2, \tag{B5}$$

with $S_{\mathrm{uds}}$ being the "unshaved" deep space measurement to determine the correction factors for $\boldsymbol{\alpha}$ pixel by pixel. The $L_o$ generated by the shaved deep space is close to zero for many pixels in the centre of the detector array, such that measurement noise prevents us from reliably determining the correction factors from the shaved measurements alone. The ozone band at $1000\,\mathrm{cm}^{-1}$ present in the unshaved deep space spectra are a sufficiently strong signal to reliably determine $\boldsymbol{\alpha}$ for all pixels of the detector. The second term is a simple ad-hoc regularisation term to make the problem numerically well-behaving. We found that $\lambda = 0.1$ works well for our purposes. In addition, we excluded all $\nu$, for which the signal generated by the calibrated unshaved deep space is smaller than $200\,\mathrm{nW\,cm}^{-2}\mathrm{sr}^{-1}\mathrm{cm}$. Otherwise, the noise error is amplified too much by the division in the cost function. We thus restricted the spectral range to $900\,\mathrm{cm}^{-1}$ to $1200\,\mathrm{cm}^{-1}$. Further, we average the spectral samples over $10\,\mathrm{cm}^{-1}$ wide bins to reduce the number of involved measurements and thus the computational effort needed for the fit.





For the function `smooth`, in principle any function capable of smoothing a 2-D grayscale image and especially of removing shot noise would be suitable (such as a median filter). We found it useful and more efficient to fit a 2-D polynomial of 20th degree to the 2-D field, remove outliers from the fit, and refit the polynomial in a second step to generate a smooth field. Thus, we get a continuous and smooth (in the mathematical sense) 2-D function with

known properties. An example of the smoothing is shown in Fig. A5: panel a shows the real part of the "unshaved" deep space spectra divided by the uncorrected gain for one wavenumber. This is effectively the sum of the calibrated spectrum and the instrumental offset. With the continuous color scale, one can already see cluster of pixels with elevated radiance values compared to the surroundings. The noisiness becomes much more apparent in Fig. A5b, where a wrapping colour scale is employed. The effect of smoothing the image with a 2-D polynomial two-step fit

is shown in Fig. A5c. The polynomial tends to overshoot at the left edge of the display, where almost all pixels are unstable due to the read-out electronics. As such, the leftmost two columns are typically discarded by default in the last processing steps.

## Appendix C: Description of the pointing system

The attitude is determined by an inertial measurement unit with laser gyroscopes from Honeywell (HG9900) that

is mounted on the gimbal yaw frame. This attitude data is combined with information from a Novatel GPS using the Kalman filtering technique to avoid, e.g., Schuler oscillations. To ensure a high stability of the pointing, a three-axis microelectromechanical (MEMS) gyroscope from Sensonor is mounted on the pitch (elevation) frame. The orientation of the gimbal frame is measured by inductive angular encoders (WMK-series) from AMO GmbH, which are robust under the harsh conditions of an open compartment even under slightly icing conditions. All these data are

fused by a stabilization control unit to provide both fast attitude changes and stabilization of the FOV. The needed agility is realized by direct-drive motors by Robodrive and independent drive controller units from Elmo Motion Control. This inertial measurement and stabilization control unit (iIMCU) was developed by iMAR Navigation GmbH, Germany.

This pointing system allows several control modes that are connected to the observation scenarios needed for

GLORIA. For atmospheric measurements with high spatial resolution, the control system switches to *Target mode* which works like a point and stare mode. During the acquisition of one interferogram, the LOS of one defined pixel is stabilized to a specific point defined in WGS84 coordinates. Then, step-by-step, the azimuth is turned by a certain angle, and the next point is stabilized. The coordinates are calculated using predefined patterns with a gimbal yaw angle step size of typically 4° to 8° and the dwell time of several seconds depending on the configured spectral

resolution of the interferometer.

For atmospheric measurements with high spectral resolution the *Altitude-Azimuth mode* is used. FOV stabilization is focused on a given tangent altitude and azimuth in the WGS84 system for a specified pixel. Contrary to the Target mode, the horizontal moving of the aircraft is not compensated, but the azimuth is kept constant during





the acquisition of one interferogram, leading to a slight horizontal smearing of the tangent point. In both modes, elevation is permanently adapted to keep the altitude of the tangent height of a specified pixel constant, even during variations of the flight altitude.

The *Nadir mode* also acts as a point and stare mode like the Target mode, but is designed to stabilize on tracing points on the ground through an aperture in the bottom of the belly pod.

Deep space calibration measurements are performed with an elevation of $10°$ in the *Angle mode* of the pointing system with stabilization of constant elevation and azimuth angles defined in WGS84.

Blackbody calibration measurements are performed in *Gimbal mode* with fixed positions of the gimbal mount in order to point to one of the two internal blackbodies.

All pointing changes are synchronized to the short breaks of interferogram acquisition when the slide in the interferometer changes direction in order to avoid scene jumps in the interferogram data.

## Appendix D:  Determining the moon position

To determine the position of the moon in relation to the instrument with high accuracy, we make use of the JPL ephemeris data in version DE421. We use the Python software package "skyfield" to access the JPL data and determine the apparent altitude and azimuth at the instrument position including relativistic effects and aberration (Rhodes, 2019). Even though the moon measurements were taken at $12\,\mathrm{km}$ altitude and above, the refraction of the atmosphere cannot be neglected, especially as some moon measurements were taken below the horizon. The skyfield package computes refraction, but only for up to -1°, a reasonable value at ground level, but insufficient for our purposes. Lacking a readily available formula for computing the refraction under the given conditions, we determined the effect of refraction by a straightforward ray-tracing algorithm given below.

Earth was assumed to be spherical, and the ray was traced in a plane with polar coordinates $(r, \theta)$ with the origin at the center of the Earth. The coefficient of refraction $n(r)$ was computed using atmospheric profiles for temperature and pressure taken from ECMWF analysis data at instrument position. Let the ray be described by $r = r_r(\theta)$, and the angle between the ray and vertical $\phi(r_r(\theta), \theta)$. We back-trace the ray from the instrument at position $(r_0, 0)$ with $\phi = \phi_0$. In the absence of refraction, the ray would be a straight line with $\phi = \phi_0 - \theta$, so for the refracted ray we can write $\phi = \phi_0 - \theta + \delta(\phi_0, r_0, n(r))$, where the angle $\delta$ represents deviation from the straight line.

$$\frac{\mathrm{d}\delta}{\mathrm{d}\theta} = \frac{\partial \delta}{\partial n} \frac{\partial n}{\partial r} \frac{\mathrm{d}r_r}{\mathrm{d}\theta} \tag{D1}$$

Snell's law gives $\mathrm{d}(n/\sin\phi) = 0$ for any point on the ray path, therefore $\mathrm{d}\delta/\mathrm{d}n = \mathrm{d}\phi/\mathrm{d}n = \tan\phi/n$. From basic geometry in polar coordinates $\partial r_r/\partial\theta = r_r/\tan\phi$. Using these two relations and equation (D1) we formulate the





back-tracing of the ray as an initial value problem

$$\begin{cases} \frac{\partial \delta}{\partial \theta} & = \frac{r}{\hat{n}+1}\frac{\partial \hat{n}}{\partial r} \\[4pt] \frac{\partial r}{\partial \theta} & = \frac{r}{\tan(\phi_0 - \theta + \delta)} \\[4pt] \delta(0) & = 0 \\[4pt] r(0) & = r_0 \end{cases} \tag{D2}$$

where the variable $\hat{n} = n - 1$ was introduced for numerical stability, as $n - 1 \ll 1$. We solve this numerically to

obtain $\phi|_{r \to \infty} \approx (\phi_0 - \theta + \delta)|_{r=2R_0}$ with $R_0$ the mean Earth radius. We then iterate the interval halving to get

moon elevation $\pi/2 - \phi_0$ as a function of the (known) astronomical elevation $\pi/2 - \phi|_{r \to \infty}$, $r_0$ and $n(r)$.

In a final step, we identified the position of the moon in the infrared images by manually matching a circle of

the expected moon size with the taken images (see Fig. 14). We expect an uncertainty in this process of about one

detector pixel for the visual identification and $0.02°$ for refraction and positioning corresponding to a total of $0.05°$

for both azimuth and elevation.

### Appendix E:  Standard atmosphere profiles

Several of our methods require standard atmospheric profiles for a range of trace gases. We typically use the profiles

collected by Remedios et al. (2007) for this purpose. Comparing forward calculations using these profiles to our

recent measurements revealed noticeable discrepancies in the order of several $\mathrm{nW\,cm^{-2}sr^{-1}cm}$ associated with

$CCl_4$, $CH_4$, $CO_2$, CFC-11, CFC-12, CFC-14, CFC-113, HCFC-22, $SF_6$, and $N_2O$. While these discrepancies can

be partly associated with the fact that a climatological standard profile will almost never fully agree with the actually

given one, we found that, especially for the upwards pointing stratospheric deep space measurements, a large part

of the discrepancy was caused by changes of atmospheric composition between the creation of the data set and the

time of measurement.

To improve the climatological profiles especially for the shaving procedure, we retrieved global monthly data

for these species from the Halocarbons & other Atmospheric Trace Species (HATS) research program (e.g., Hu

et al., 2016; Montzka et al., 2021, 1996; Elkins and Dutton, 2009; Montzka et al., 2009; Conway et al., 1994;

Dlugokencky et al., 1994). We further smoothed the data by computing a running mean over twelve consecutive

890    months and fully disregard differences between the hemispheres for simplicity. For HCFC-22 and CFC-14, no such

globally averaged product was available at the time; instead measurement data from several individual stations was

given. We computed a crude global average over the available stations and also performed a running yearly mean to

derive a similar product as given for the other species.

To update the profiles of Remedios et al. (2007), we simply scale the full profile (and the associated standard

deviation) with a multiplicative factor derived from the quotient between the value of the HATS data and the ground



level value of the profile. This improved the fit of radiances and reduced visible residuals by these interfering species and thus reduced the systematic error induced by emissions from these trace gases.

## Appendix F: Level 2 processing

### F1 line of sight/temperature retrieval

To determining the line of sight from measured data with a spectral resolution of $0.2\,\mathrm{cm}^{-1}$, we assume that temperature values supplied by ECMWF analysis data are close to real values. In effect, we perform a standard retrieval keeping temperature at ECMWF values and only vary the elevation correction value until the simulated measurements fit the measured ones best. A very rough error estimate of the approach shows that with gradients of $\approx 10\,\mathrm{K\,km}^{-1}$ a systematic 1 K error in ECMWF would only cause an error in final pointing data of about 100 m.

The situation becomes even better when profiles contain both tropospheric and stratospheric sections with temperature gradients of opposite sign. Still wrongly represented tropopauses, gravity waves, or other non-uniform biases in temperature impact the method. We thus do (generally) not trust in individual values, but in the average over a whole measurement flight after filtering short profiles with high uncertainty.

The retrieval itself leverages $CO_2$ emission lines around $950\,\mathrm{cm}^{-1}$, where only few other gases have interfering

emissions. Table 6 shows the used spectral regions. These were selected to avoid strong emission features by, e.g., $CFC-12$, $O_3$, $H_2O$ and $SF_6$ and cover as many $CO_2$ lines as feasible. The retrieval uses the trace gasses of $CO_2$, $CFC-12$, and $SF_6$ with climatological values (see Appendix E). For $H_2O$, we use water vapour as given in the ECMWF analysis data, even though the impact of this is small. In addition to the line of sight, an extinction profile is also retrieved to capture both atmospheric aerosol and small calibration errors.

To derive temperature instead, we simply revert the roles of line of sight and temperature. We use smoothed ECMWF temperatures as a priori and initial guess (but the choice of a priori typically does not affect the results, only the convergence speed). In addition to temperature, this setup also derives PAN as secondary target to remove its (small) influence in polluted air.

### F2 $O_3$ retrieval

The ozone retrieval discussed briefly in Sect. 6 is a simplified version of the retrieval described by (Ungermann et al., 2015). It uses the four spectral regions listed in Table 7, each averaged to a single radiance. The temperature and PAN values derived by the temperature retrieval of Sect. F1 are used as well as ECMWF analysis water vapour. The retrieval derives only $O_3$ volume mixing ratios and a single extinction profile. Climatological values are employed for the trace gases $CO_2$, CFC-12, CFC-113, $NH_3$, and $SF_6$ (see Appendix E).





*Author contributions.* JU and AK developed or contributed to all work in this paper and wrote most of it. GM constructed and described the pointing system and its calibration. IB contributed the analysis of parasitic images. FFV contributed to the instrument description and NESR and PSF analysis. SJ contributed to the line of sight analysis and level 2 error diagnosis. LK contributed the refraction correction for moon measurements. TN helped with all effects related to GLORIA electronics. All authors reviewed the whole paper and provided many corrections and suggestions. All authors supported the development and/or 930 operation of GLORIA in the field.

*Competing interests.* The authors declare that they have no conflict of interest.

*Acknowledgements.* Atmospheric research with HALO is supported by the Priority Programme SPP 1294 of the Deutsche Forschungsgemeinschaft (DFG). This work was partly funded by the German Ministry for Education and Research under grant 935 01 LG 1907 (project WASCLIM) in the frame of the Role of the Middle Atmosphere in Climate (ROMIC)-program. The European Centre for Medium-Range Weather Forecasts (ECMWF) is acknowledged for meteorological data support. We especially thank the full GLORIA team, including the institutes ZEA-1, ZEA-2 at Forschungszentrum Jülich and the Institute for Data Processing and Electronics at the Karlsruhe Institute of Technology, for their great work during the campaigns on which all the data in this paper are based. We would also like to thank the pilots and the ground support team at the Flight Experiments 940 facility of the Deutsches Zentrum für Luft- und Raumfahrt (DLR-FX). We thank Albert Adibekyan, Christian Monte and Max Reiniger from PTB, Germany for support on the characterization of the blackbodies and the Germanium entrance window in the frame of the EMPIR Project 16ENV03, "Metrology for Earth Observation and Climate 3" (MetEOC-3). We really appreciate the competent collaboration with iMAR Navigation and their very good support whenever it was needed. Data was provided by the NOAA Global Monitoring Laboratory, Boulder, Colorado, USA (https://esrl.noaa.gov/).





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



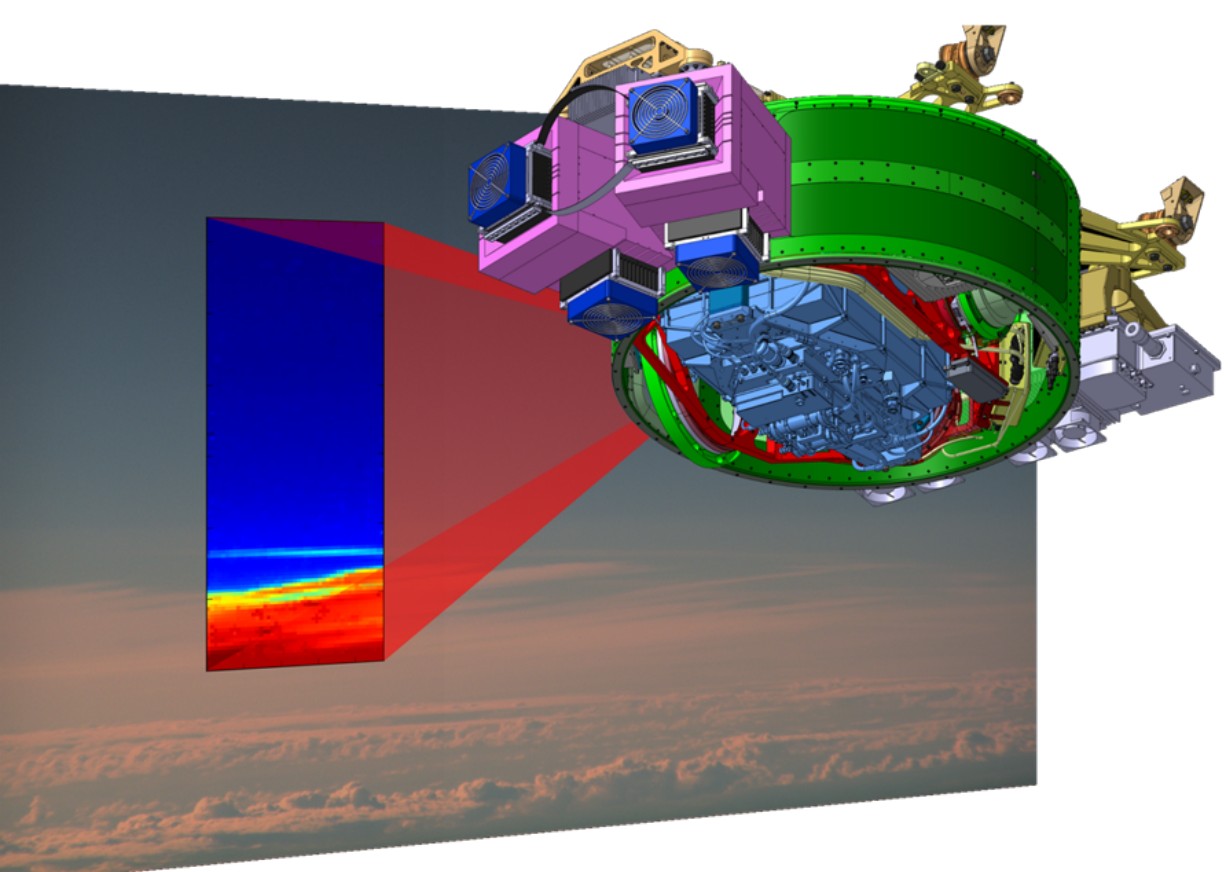

**Figure 1.** Schematic of the instrument and the measurement principle. Depicted is an overlay of an IR and an visual image together with a technical drawing of the instrument.



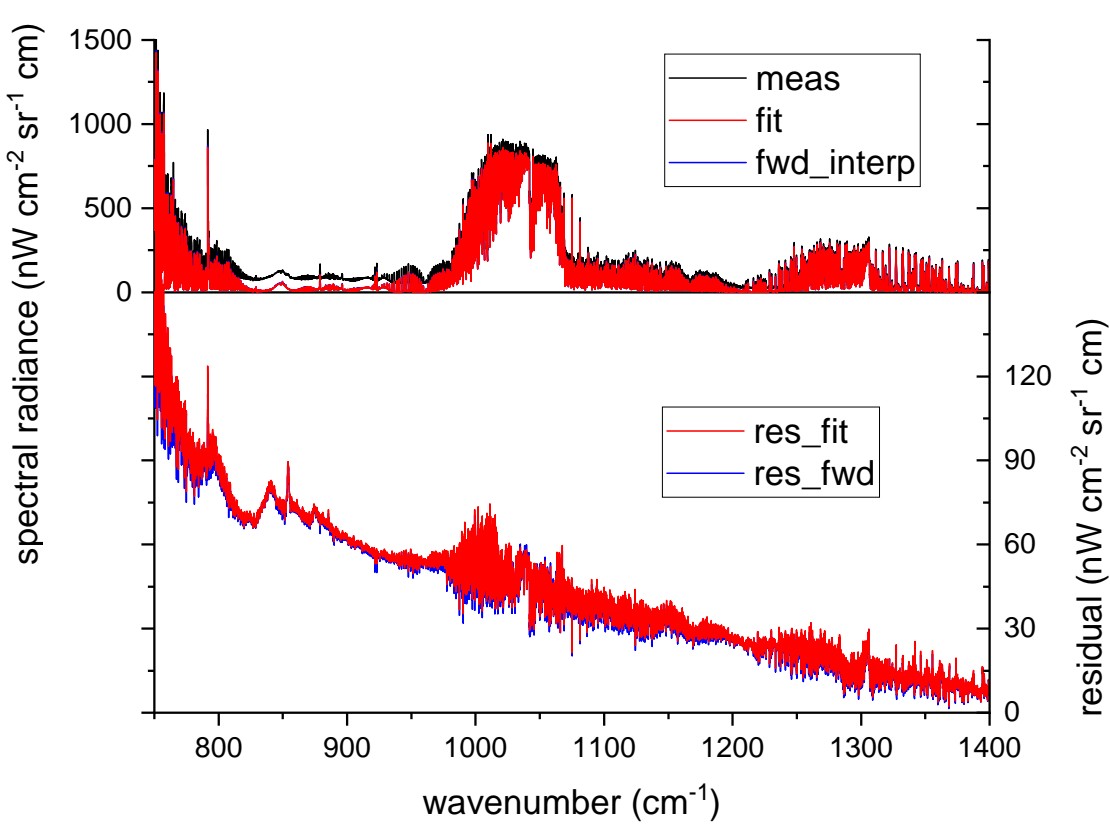

**Figure 2.** Original measured spectrum without subtraction of the Planck function (black), forward calculated spectrum from the fit results (red), and linearly interpolated spectrum from the forward calculations for the lowermost and uppermost detector row (blue, hidden by the red line). The lower panel shows the residuals (measured minus fit in red and measured minus interpolated forward calculation in blue), enlarged by a factor of 10, demonstrating the very small difference between fitted and forward calculated interpolated spectrum.



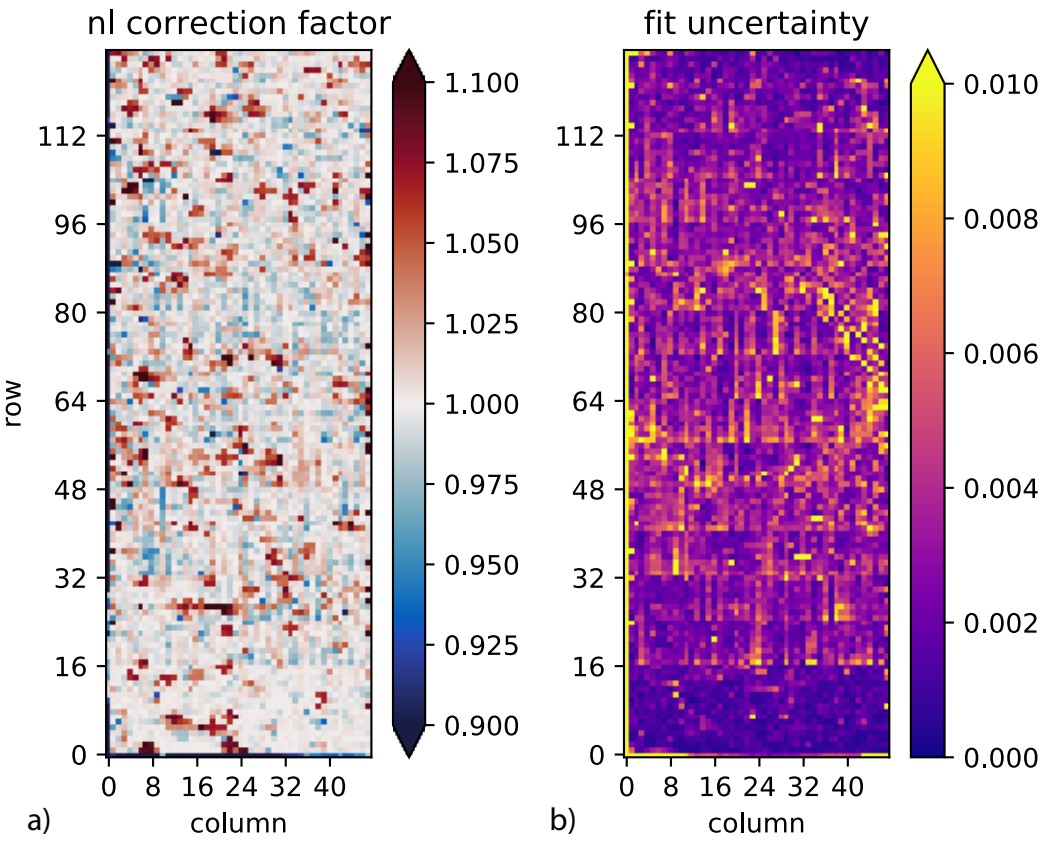

**Figure 3.** The correction factors for blackbody raw spectra derived for a blackbody-deep space calibration sequence of the WISE campaign/flight 16. Panel (**a**): the correction factors. Panel (**b**): the uncertainties derived from the least squares fit.



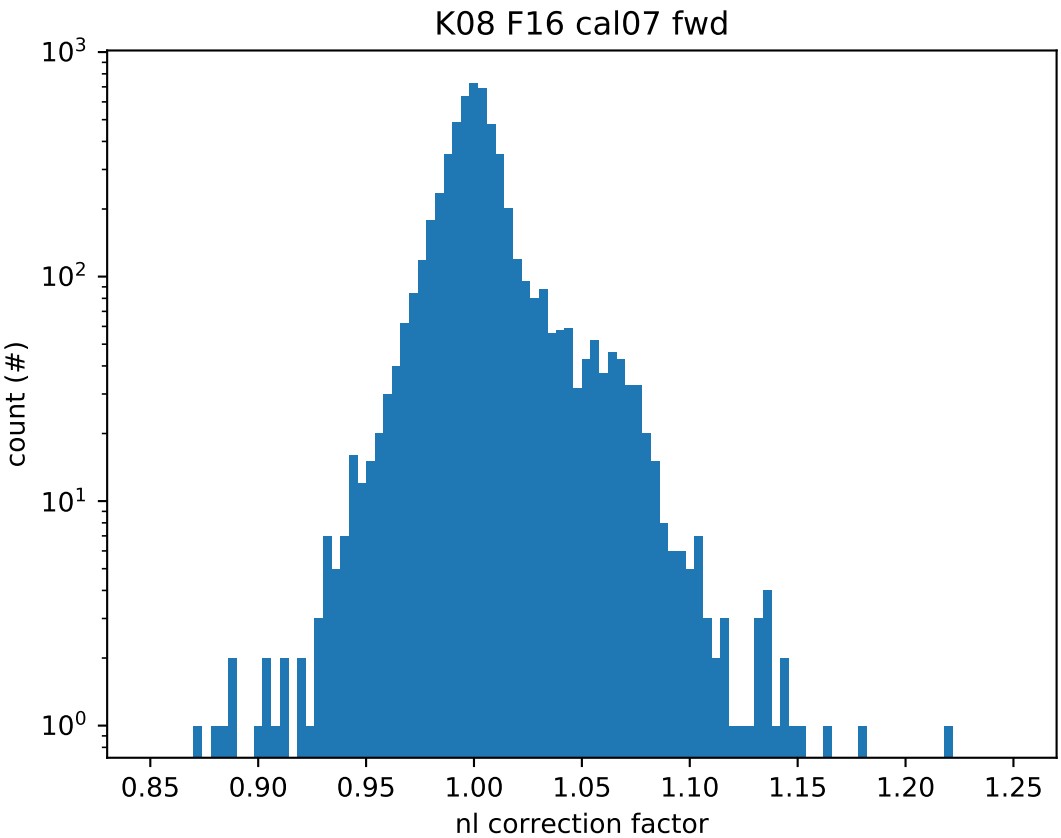

**Figure 4.** The frequencies of different non-linearity correction factors derived from one calibration sequence in a logarithmic scale.



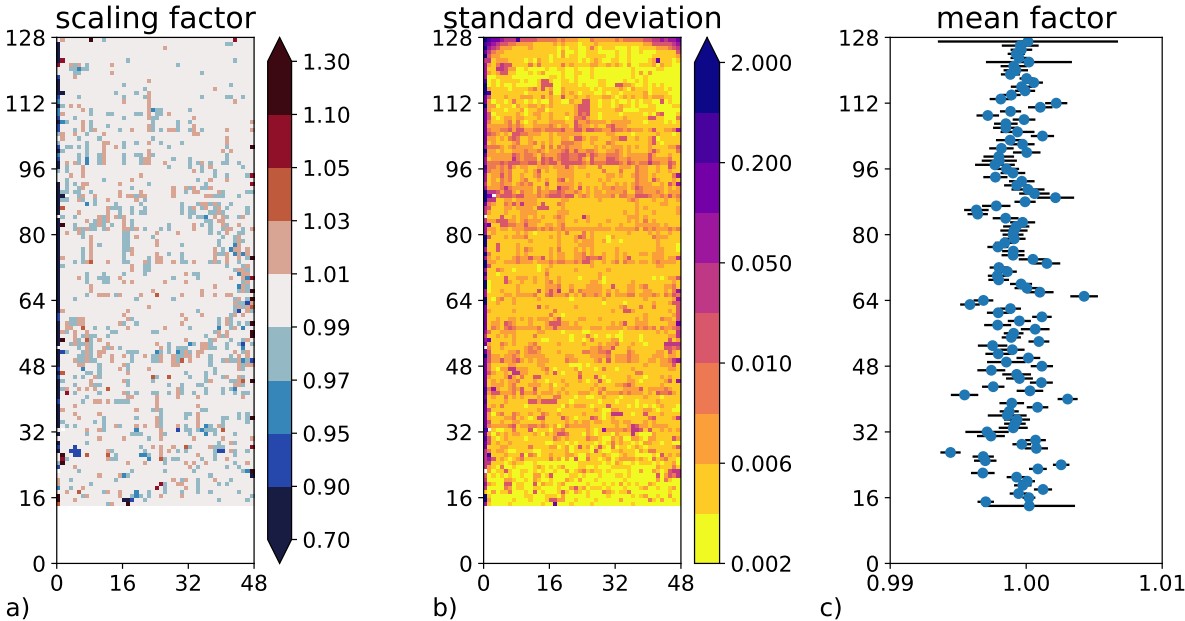

**Figure 5.** Panel **(a)**: the estimated error in gain derived from homogeneity of 158 atmospheric spectra. Panel **(b)**: the corresponding standard deviation of the variability over the same atmospheric spectra. Panel **(c)**: the estimated error in gain after averaging over all pixels in a row (excluding the leftmost two and rightmost columns). All atmospheric spectra were cloudy in the lowermost rows, precluding an analysis.





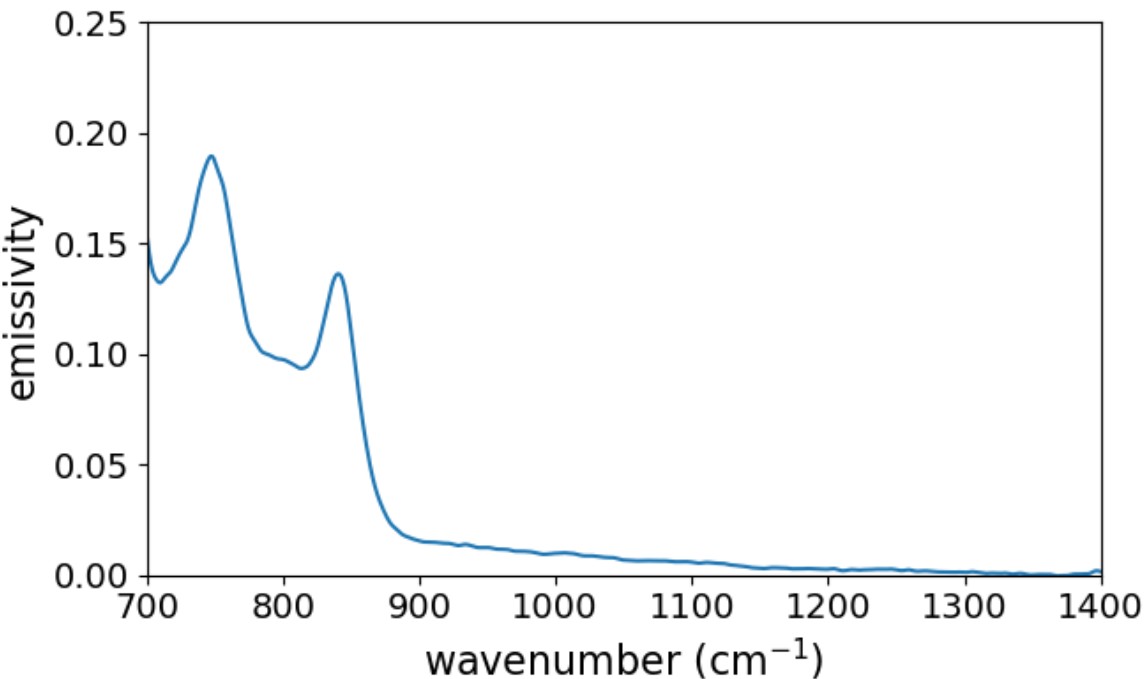

**Figure 6.** The spectral emissivity of the outer window.





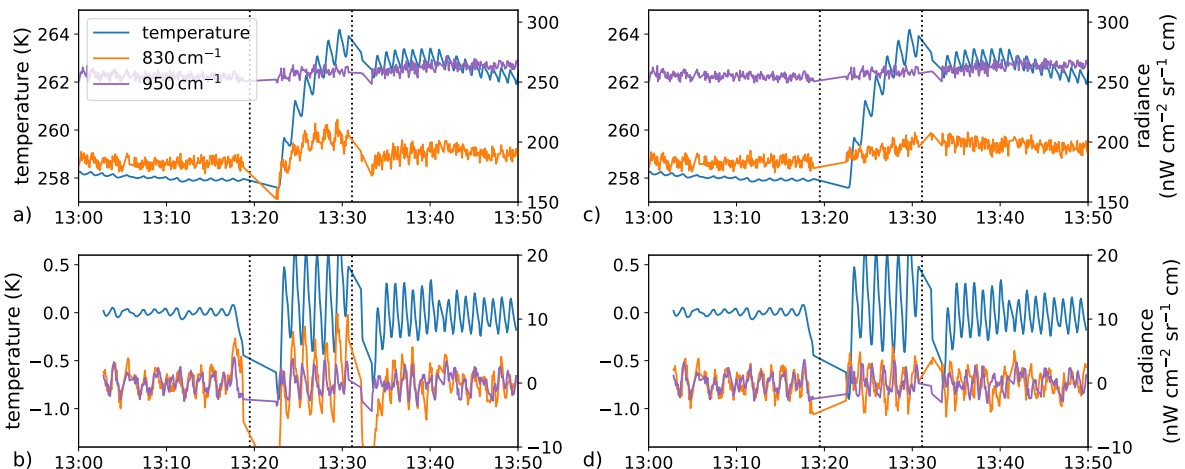

**Figure 7.** Averaged radiances for part of flight 13 of the WISE campaign during a tomographic measurement pattern for (left) original and (right) corrected radiances. Outer window temperature is given in blue. Shown in orange and purple are radiances for central wavenumbers $830\,\text{cm}^{-1}$ and $950\,\text{cm}^{-1}$, averaged over six wavenumbers each and over rows 68 to 72 (except for pixels flagged as faulty). The times of calibration measurements are shown as vertical dotted lines. Panel **(a)**: radiance values without window correction. Panel **(b)**: only the high-frequency part of the signal. Panels **(c)**, **(d)**: the same radiances with window correction enabled, correspondingly.

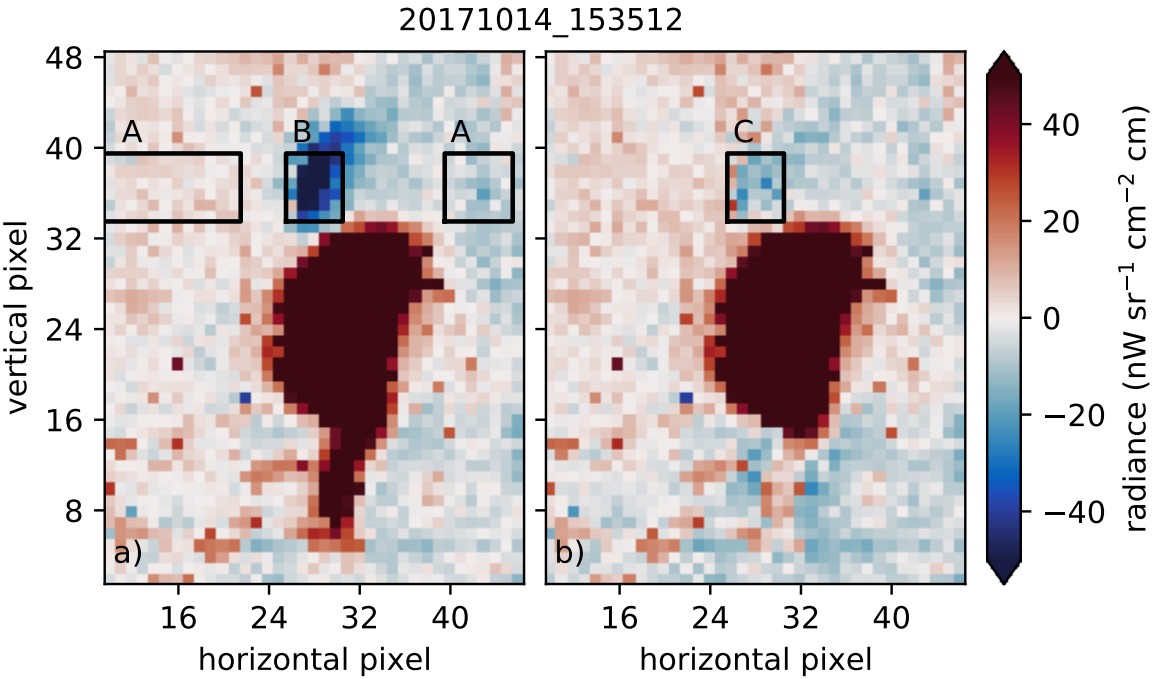

**Figure 8.** Panel **(a)** shows the average radiance of a moon image from 14 October 2017, with two strong parasitic images above and below. Panel **(b)** shows the same image after the parasitic image correction was applied. From both images the same background was subtracted, computed for each row individually from the median over all pixels outside the columns containing moon affected pixels. The black rectangles indicate regions related to the spectra shown in Fig. 9.

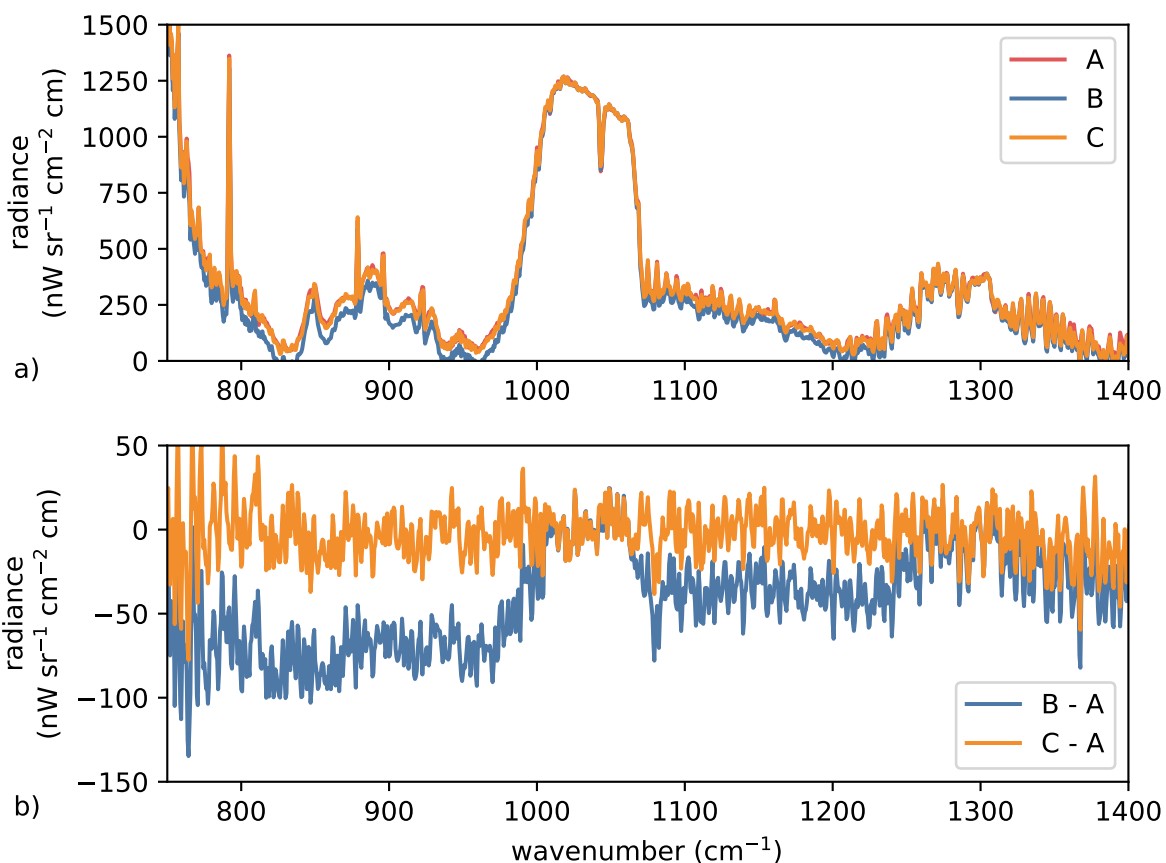

**Figure 9.** Panel **(a)**: spectra averaged over all pixels marked by black rectangles in Fig. 8. 'A' refers to all pixels in Fig 8a in the two outer rectangles, whereas 'B' refers to all pixels within the center rectangle. 'C' refers to all pixels in the center rectangle of Fig. 8b (The 'A' and 'C' spectra are very similar). Panel **(b)**: differences over the spectra of panel (a) to highlight the differences.



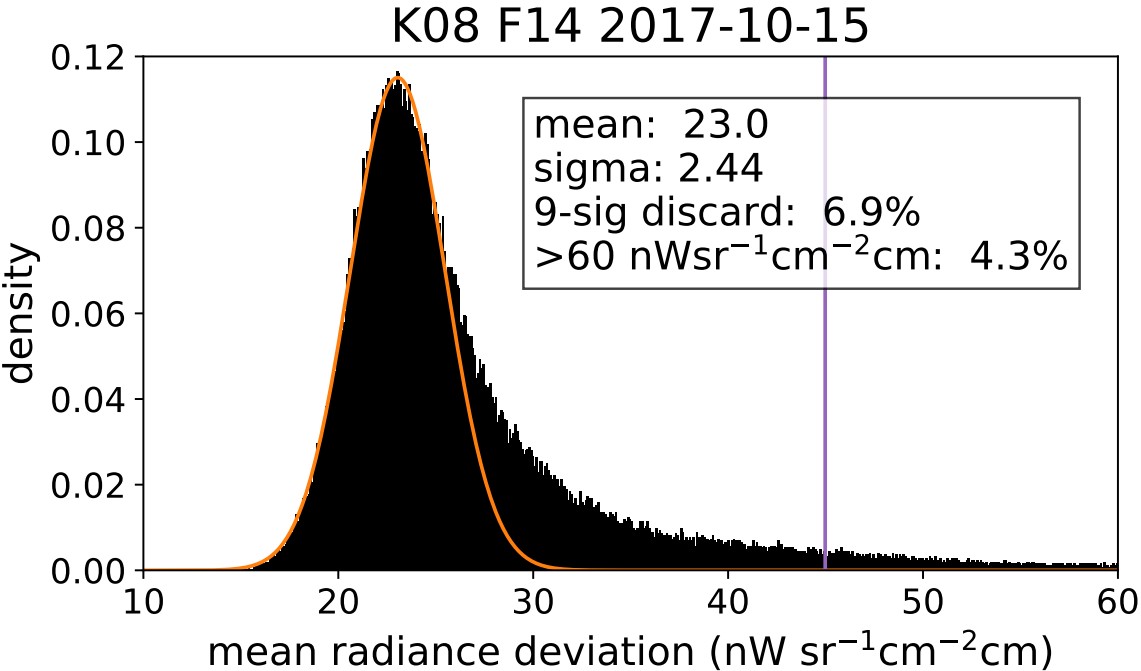

**Figure 10.** Distribution of spectrally averaged absolute differences to the horizontal median value for the 14th flight of the WISE campaign. A Gaussian (in orange) curve that was fitted to the left hand side of the distribution. The vertical bar (purple) shows the chosen 9-sigma threshold. 4.3% of all values lie outside of this plot.





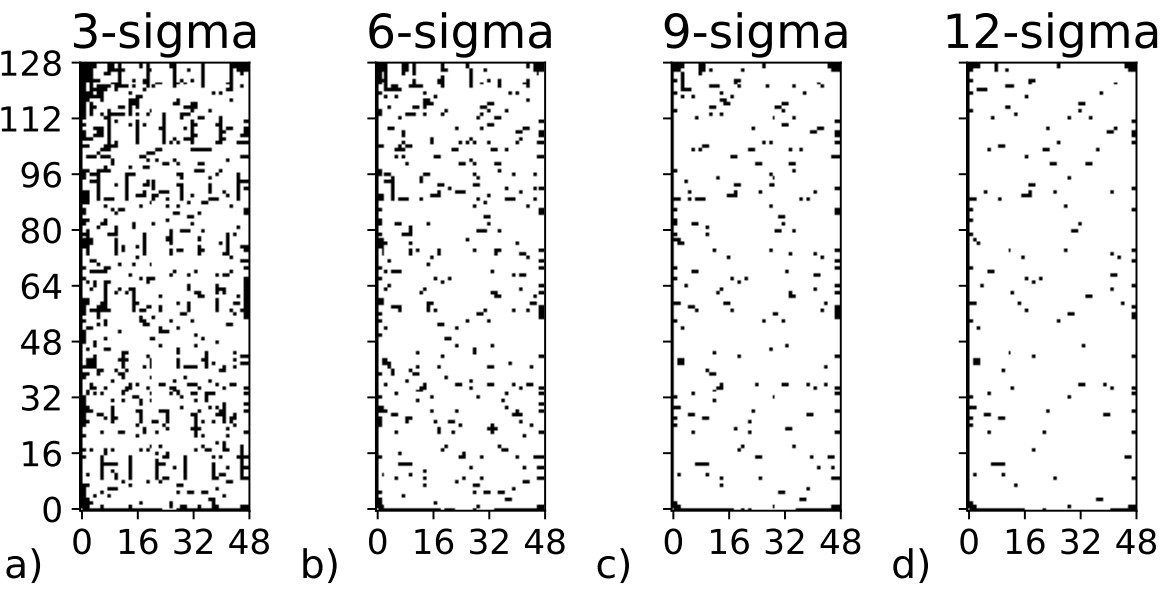

**Figure 11.** Different bad pixel masks for the 14th flight of the WISE campaign, bad pixels in black. Panel **(a)**: for a 3-sigma threshold (19.0% filtered). Panel **(b)**: for a 6-sigma threshold (10.4% filtered). Panel **(c)**: for a 9-sigma threshold (6.9% filtered). Panel **(d)**: for a 12-sigma threshold (5.2% filtered).





**Figure 12.** Pointing calibration optics with an off-axis paraboloid mirror which is delivering parallel beams with a broadband infrared light source in front of the GLORIA instrument (golden) integrated in the belly pod of HALO in Oberpfaffenhofen. In between the theodolite during the adjusting phase of the beams.

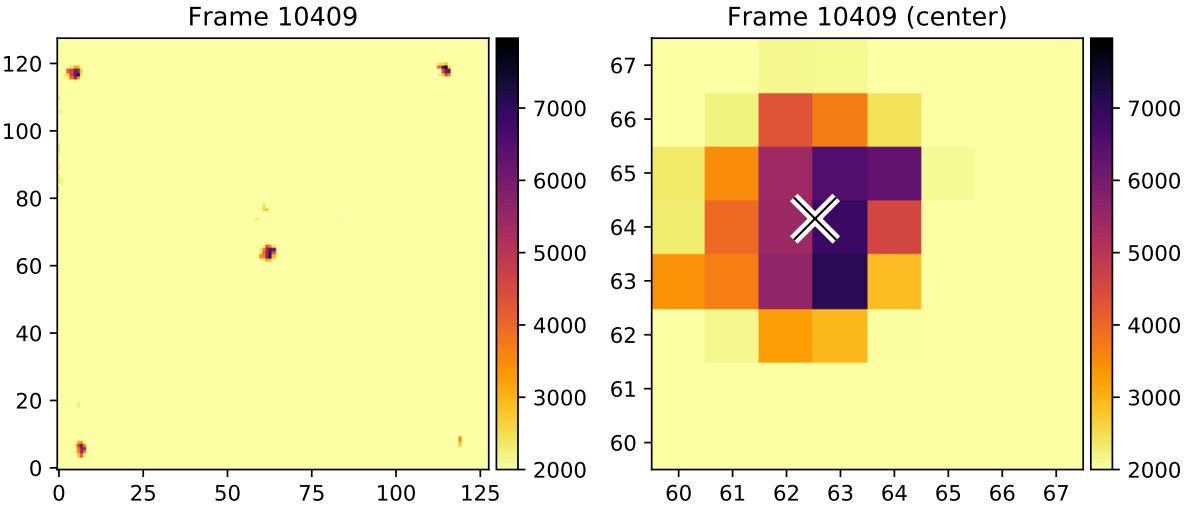

**Figure 13.** Measurements of the five laser beams of a theodolite. Panel **(a)** shows the full detector, being read out with $128 \times 128$ pixels for this measurement. Panel **(b)** shows a close up of the centre beam that highlights the 0° pitch / 90° azimuth position.





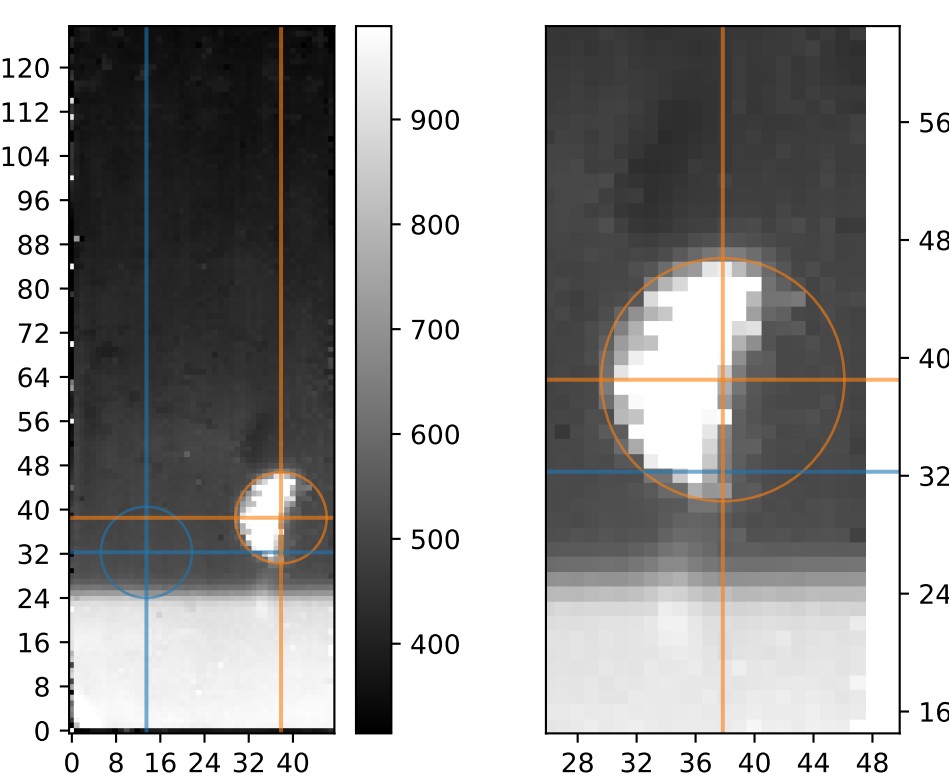

**Figure 14.** Example of the determination of moon position for one cube measured at 2017–10–14T15:37:58Z. Panel (**a**) shows the full image while panel (**b**) shows the moon in an enlarged fashion. The expected moon position is shown in blue, the manually determined actual one in orange. Here, the difference between expected and actual moon position is 0.78° horizontally and 0.2° vertically.



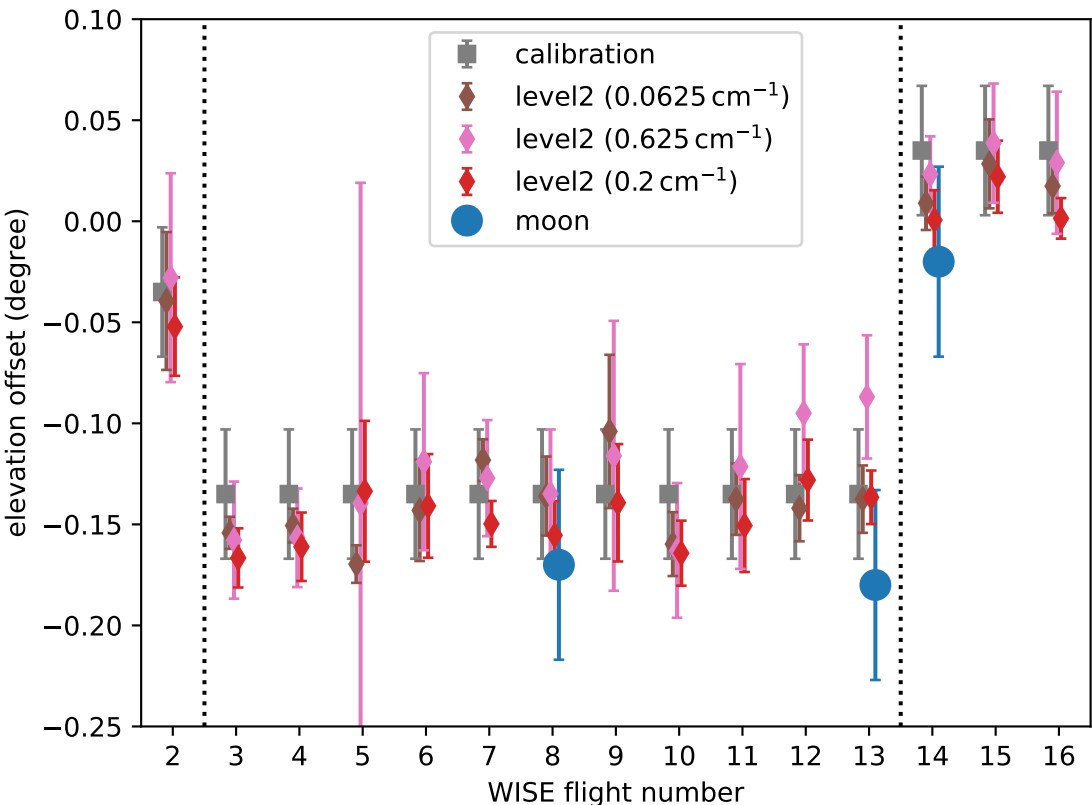

**Figure 15.** Values derived for absolute elevation offset during the WISE campaign. Blue are offset determined from moon measurements, grey are the values derived from calibration measurements pre- and post-campaign (pre-campaign for the first two flights, post-campaign for the remaining). Pink, red and brown are elevation angles as determined from level 2 processing and ECMWF temperature data. The leftmost dotted line marks a change of the inertial navigation system, which invalidated the attitude calibration. The right-most dotted line marks a change of employed elevation angle offset correction, where we corrected the pointing according to our then best level 2 data.





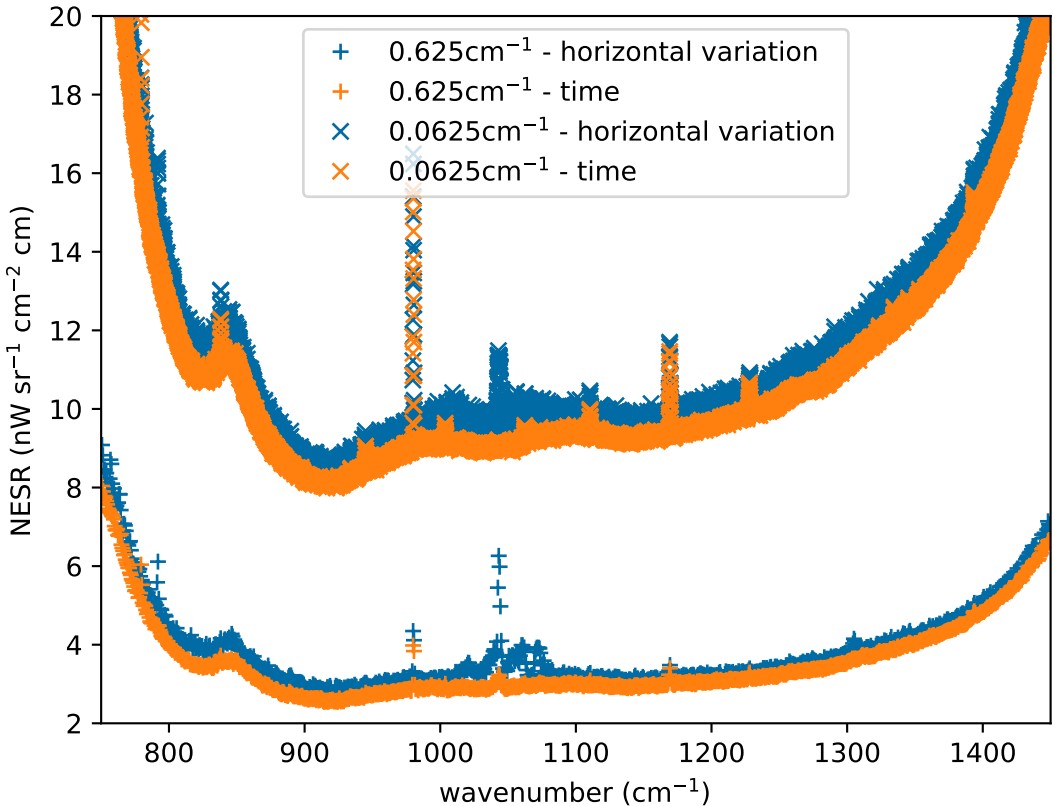

**Figure 16.** Exemplary NESR estimates from a sequence of deep space measurements in two different spectral resolutions and using two different methods.





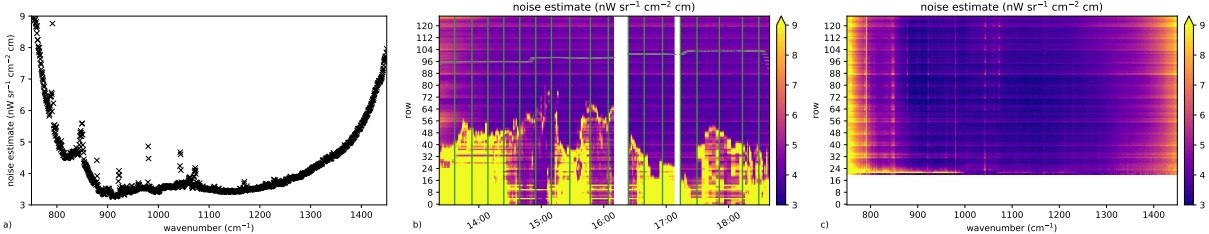

**Figure 17.** Exemplary NESR analysis of atmospheric spectra for the WISE flight of 15th October 2017. Panel **(a)** shows a single NESR spectra averaged over time and pixels. Panel **(b)** shows NESR spectrally averaged. Panel **(c)** shows NESR averaged over time.



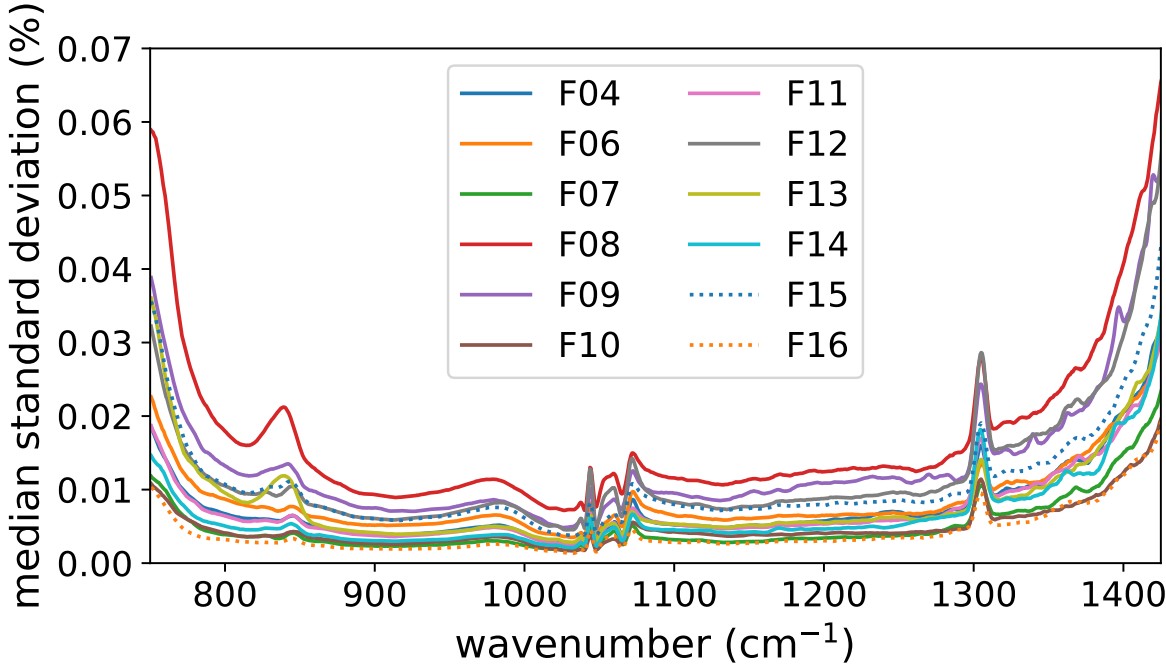

**Figure 18.** Uncertainty in gain magnitude. Depicted is the standard deviation computed over the individual gain magnitudes from all calibration times of the given flights from the WISE campaign.





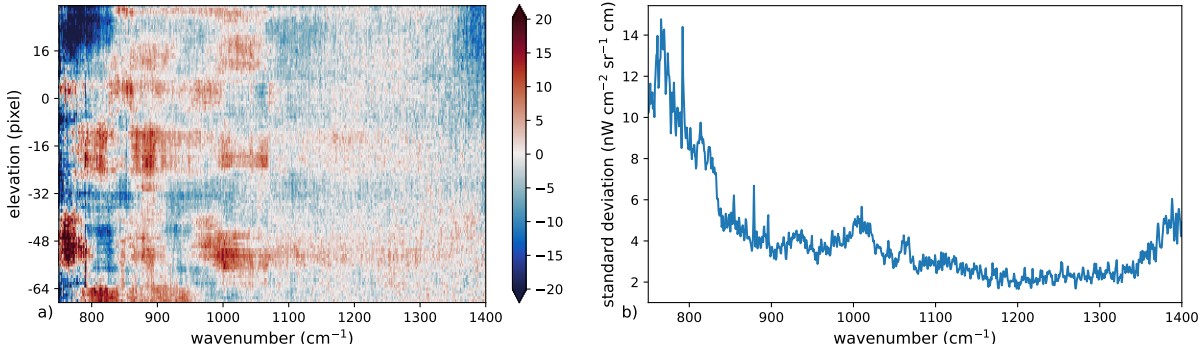

**Figure 19.** Difference (panel **(a)**) between calibrated radiances measuring the same air mass at different elevation angles and standard deviation thereof **(b)** The measurements were taken during the POLSTRACC campaign on the 2nd February 2016 at 22:00 UTC alternating between -0.38° and -1.00° elevation.





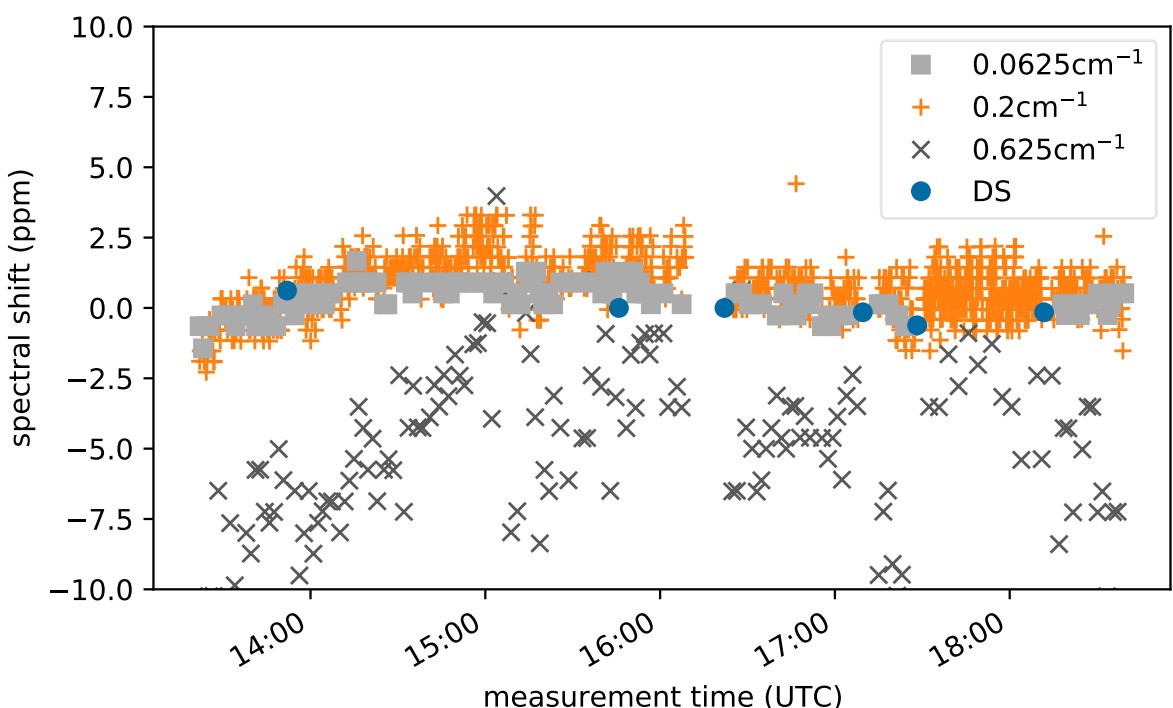

**Figure 20.** Exemplary spectral shift analysis of atmospheric spectra for the WISE flight of 15th October 2017. The shift derived from atmospheric measurements processed in different spectral resolutions is depicted.

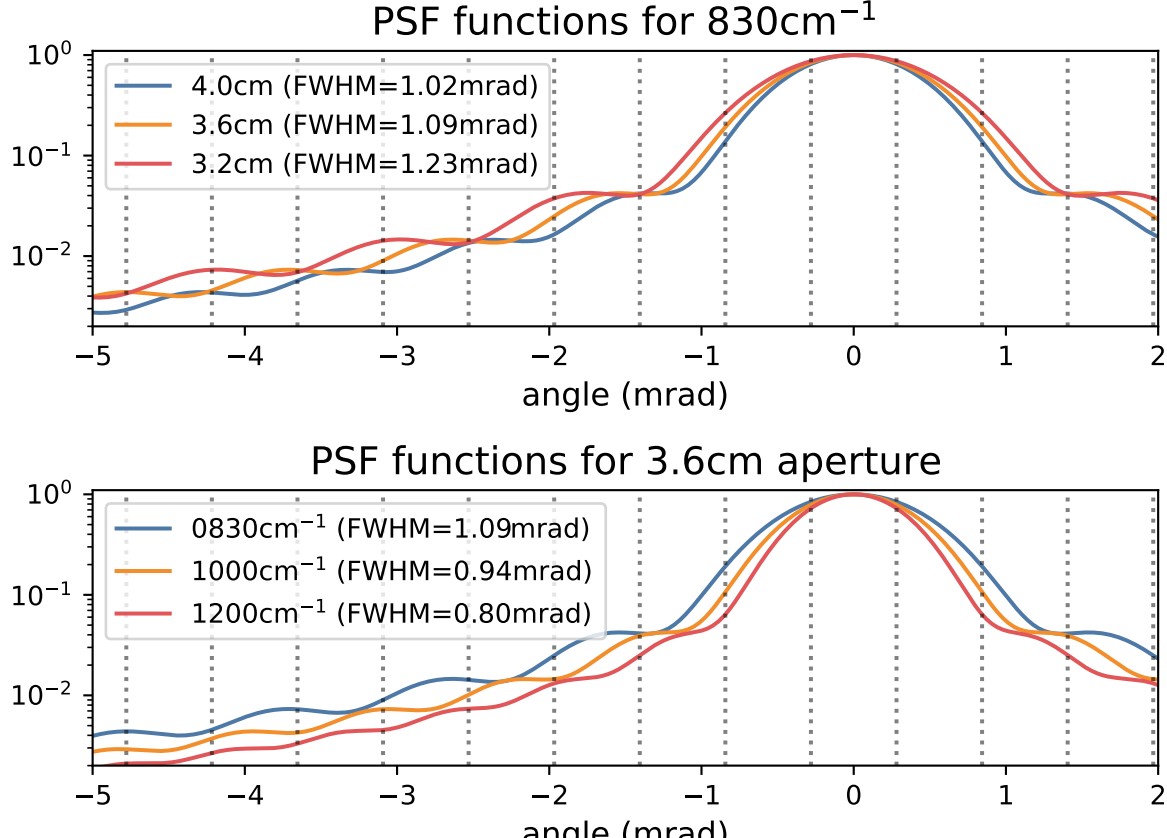

**Figure 21.** The impact of aperture (upper panel) and wavenumber (lower panel) on the theoretical point spread function.



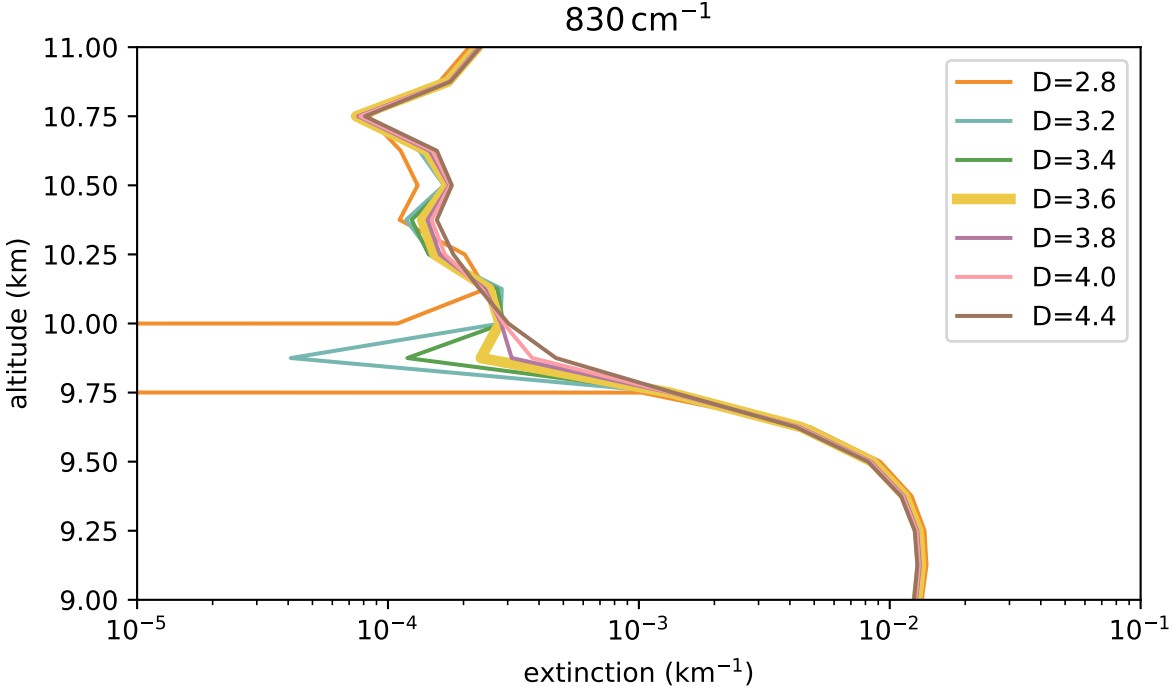

**Figure 22.** Exemplary derived extinction profiles employing point spread functions with differing apertures.



**Table 1.** Typical operating modes of GLORIA. Actual optical path difference is typically longer than the nominal one. Temporal resolution is given approximately as it partially depends on the configuration of the instrument.

| name | nominal optical path difference (cm) | spectral resolution ($cm^{-1}$) | temporal resolution (s) |
| --- | --- | --- | --- |
| high spatial resolution mode (dynamics mode) | 0.8 | 0.625 | 2.5 |
| intermediate resolution mode | 2.5 | 0.2 | 5.0 |
| high spectral resolution mode (chemistry mode) | 8.0 | 0.0625 | 13.5 |



**Table 2.** Microwindows and fit parameters used in the different iterations of the fitting process. Gases considered in the forward calculation using the a priori profiles are $H_2O$, $CO_2$, $O_3$, $N_2O$, $CH_4$, $O_2$, $SO_2$, $NO_2$, $HNO_3$, $ClO$, $OCS$, $HOCl$, $N_2$, $HCN$, $CH_3Cl$, $H_2O_2$, $C_2H_2$, $COF_2$, $CFC-11$, $CFC-12$, $HCFC-22$, $SF_6$, $CFC-14$, $CCl_4$, $CFC-113$, $CFC-114$, $N_2O_5$, $ClONO_2$, $HNO_4$. Furthermore, the continuum of $O_2$ and $H_2O$ is considered. For further details, see text.

| name | microwindow | fit parameters |
|---|---|---|
| offset fit | 815 – 820 | shift, offset, scale |
| | 937 – 941 | shift, offset, scale |
| | 950 – 954 | shift, offset, scale |
| | 1214 – 1218 | shift, offset, scale |
| | 1402 – 1406 | shift, offset, scale |
| broadband fit | 750 – 1400 | T, $H_2O$, $CO_2$, $O_3$, $N_2O$, $CH_4$, $HNO_3$, CFC-11, CFC-12 |
| gas fits 1 | 935 – 970 | $CO_2$ |
| | 1177 – 1190 | $N_2O$ (v0) |
| | 832 – 858 | CFC-11 |
| | 910 – 935 | CFC-12 |
| gas fits 2 | 970 – 1065 | $O_3$ (v1) |
| | 860 – 910 | $HNO_3$ |
| | 828 – 830 | HCFC-22 |
| | 779.875 – 780.5 | $ClONO_2$ |
| | 940 – 955 | $SF_6$ |
| gas fits 3 | 1065 – 1170 | $O_3$ (v2) |
| | 1205 – 1285 | $N_2O$, $CH_4$ (v1) |
| | 1306.5 – 1400 | $N_2O$, $CH_4$ (v2) |





**Table 3.** Average noise value of a $0.625\,\mathrm{cm}^{-1}$ spectral sample and maximum noise of a superpixel averaged over all spectral $0.625\,\mathrm{cm}^{-1}$ samples for different bad pixel masks. The smaller the $n$ in $n$-sigma, the more pixels will be filtered.

| mask | average | maximum |
|---|---|---|
| no | $8.96\,\mathrm{nW\,cm}^{-2}\mathrm{sr}^{-1}\mathrm{cm}$ | $21.23\,\mathrm{nW\,cm}^{-2}\mathrm{sr}^{-1}\mathrm{cm}$ |
| 15-sigma | $4.38\,\mathrm{nW\,cm}^{-2}\mathrm{sr}^{-1}\mathrm{cm}$ | $6.24\,\mathrm{nW\,cm}^{-2}\mathrm{sr}^{-1}\mathrm{cm}$ |
| 12-sigma | $4.36\,\mathrm{nW\,cm}^{-2}\mathrm{sr}^{-1}\mathrm{cm}$ | $6.28\,\mathrm{nW\,cm}^{-2}\mathrm{sr}^{-1}\mathrm{cm}$ |
| 9-sigma | $4.33\,\mathrm{nW\,cm}^{-2}\mathrm{sr}^{-1}\mathrm{cm}$ | $6.16\,\mathrm{nW\,cm}^{-2}\mathrm{sr}^{-1}\mathrm{cm}$ |
| 6-sigma | $4.30\,\mathrm{nW\,cm}^{-2}\mathrm{sr}^{-1}\mathrm{cm}$ | $5.26\,\mathrm{nW\,cm}^{-2}\mathrm{sr}^{-1}\mathrm{cm}$ |
| 3-sigma | $4.37\,\mathrm{nW\,cm}^{-2}\mathrm{sr}^{-1}\mathrm{cm}$ | $5.64\,\mathrm{nW\,cm}^{-2}\mathrm{sr}^{-1}\mathrm{cm}$ |



**Table 4.** This table collects (simplified) estimates for the examined error sources.

| | |
|---|---|
| NESR ($880\,\mathrm{cm}^{-1}$–$1300\,\mathrm{cm}^{-1}$, $0.625\,\mathrm{cm}^{-1}$) | $5\,\mathrm{nW\,cm}^{-2}\mathrm{sr}^{-1}\mathrm{cm}$ |
| NESR (otherwise, $0.625\,\mathrm{cm}^{-1}$) | $8\,\mathrm{nW\,cm}^{-2}\mathrm{sr}^{-1}\mathrm{cm}$ |
| deep space shaving | $20\,\mathrm{nW\,cm}^{-2}\mathrm{sr}^{-1}\mathrm{cm}$ |
| detector non-linearity | 0.2% (of gain) |
| total gain | 1% |
| outer window emission | $<1\,\mathrm{nW\,cm}^{-2}\mathrm{sr}^{-1}\mathrm{cm}$ |
| instrument pointing | 0.032° |
| PSF width | 10% |
| offset accuracy | $10\,\mathrm{nW\,cm}^{-2}\mathrm{sr}^{-1}\mathrm{cm}$ |
| spectral accuracy | 5 ppm |



**Table 5.** This table collects the impact of the specified error source on two exemplary level 2 products.

| error source | temperature | O$_3$ |
|---|---|---|
| NESR | -0.00K±0.12K | 0.08%±1.84% |
| deep space shaving | 0.27K±0.06K | 0.72%±1.82% |
| offset accuracy | -0.02K±0.08K | 0.18%±2.61% |
| detector non-linearity | -0.01K±0.05K | 0.05%±0.99% |
| total gain | -0.25K±0.07K | -1.45%±1.83% |
| instrument pointing | -1.14K±0.28K | 3.57%±4.08% |
| PSF width | 0.02K±0.03K | 0.05%±0.88% |
| spectral accuracy | 0.01K±0.02K | -0.13%±0.52% |





**Table 6.** Spectral regions used for the line of sight and temperature retrievals ($0.2\,\mathrm{cm}^{-1}$ resolution).

| |
|---|
| $936.8\,\mathrm{cm}^{-1}$–$940.4\,\mathrm{cm}^{-1}$ |
| $941.4\,\mathrm{cm}^{-1}$–$944.2\,\mathrm{cm}^{-1}$ |
| $951.2\,\mathrm{cm}^{-1}$–$952.8\,\mathrm{cm}^{-1}$ |
| $956.2\,\mathrm{cm}^{-1}$–$958.2\,\mathrm{cm}^{-1}$ |





**Table 7.** Spectral regions used for the $O_3$ retrieval ($0.2\,\mathrm{cm}^{-1}$ resolution).

| |
|---|
| $936.8\,\mathrm{cm}^{-1}$–$940.4\,\mathrm{cm}^{-1}$ |
| $956.8\,\mathrm{cm}^{-1}$–$962.4\,\mathrm{cm}^{-1}$ |
| $980.0\,\mathrm{cm}^{-1}$–$984.2\,\mathrm{cm}^{-1}$ |
| $992.6\,\mathrm{cm}^{-1}$–$997.4\,\mathrm{cm}^{-1}$ |
| $1000.6\,\mathrm{cm}^{-1}$–$1006.2\,\mathrm{cm}^{-1}$ |



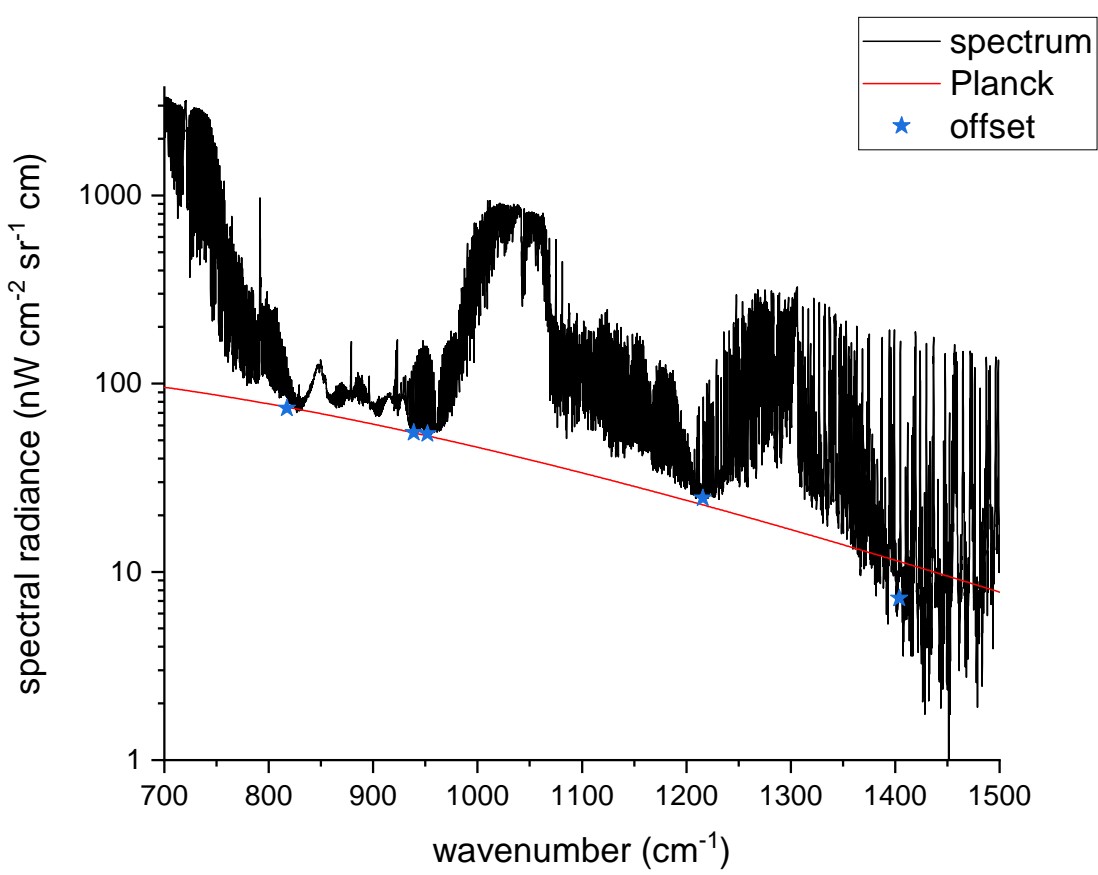

**Figure A1.** Median calibrated deep space spectrum together with the offset values at 5 spectral points and a Planck function fitted to the offset points. It was taken during the SouthTRAC campaign on 8 September 2019 at 7:53 UTC. Please note the logarithmic scale.



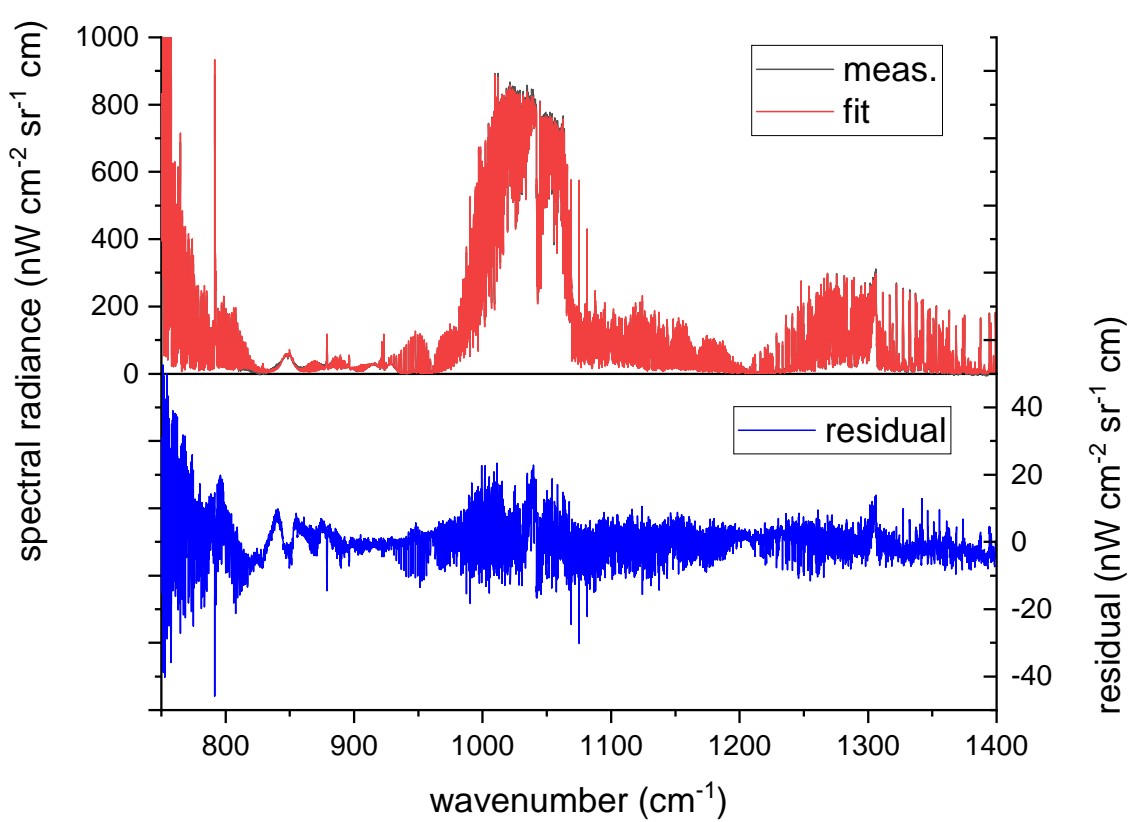

**Figure A2.** Upper panel: the measurement after subtraction of the Planck function and fit result of the broadband fit. Lower panel: the residual (measurement-fit) on the right ordinate, enlarged by a factor of 10.

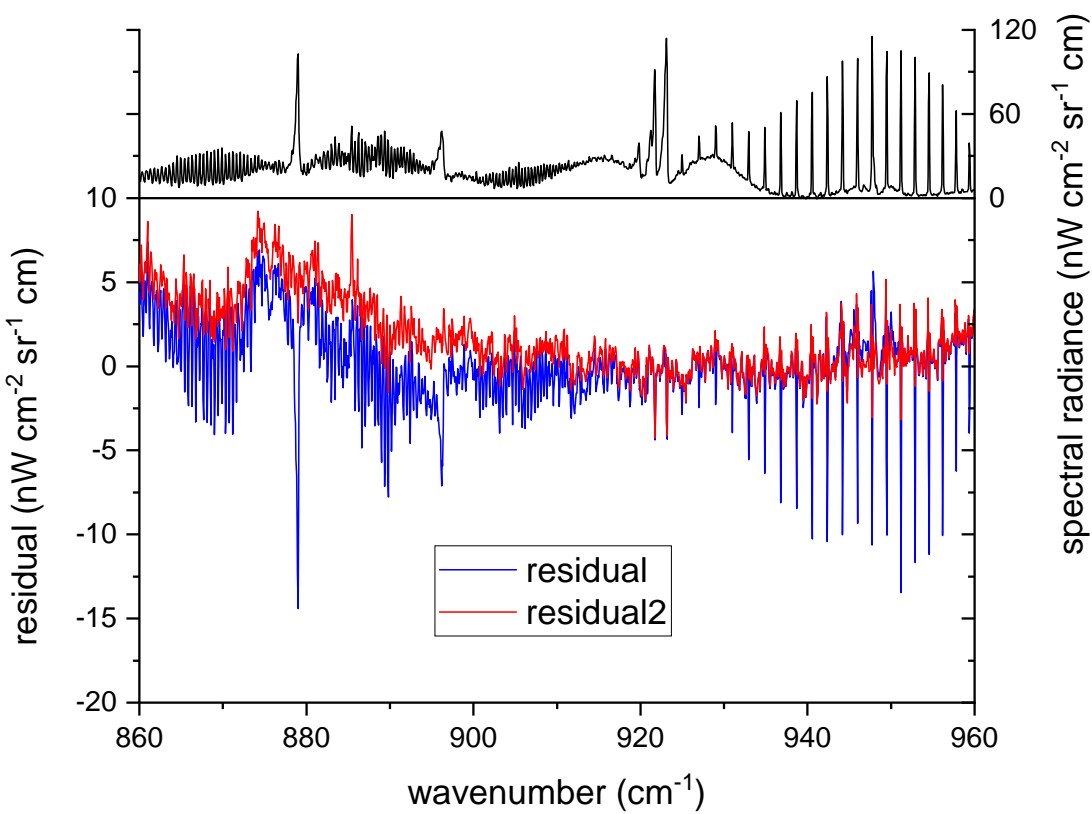

**Figure A3.** Residual in the spectral range of 860 to 960 $cm^{-1}$ after the broadband fit (blue) and after the fit of selected gases in individual microwindows (red). The upper panel shows the measured spectrum on the right ordinate, reduced by a factor of 12. The spectral signatures of $HNO_3$ and $CO_2$ are clearly reduced after the second fit, as well as $SF_6$ around 948 $cm^{-1}$.





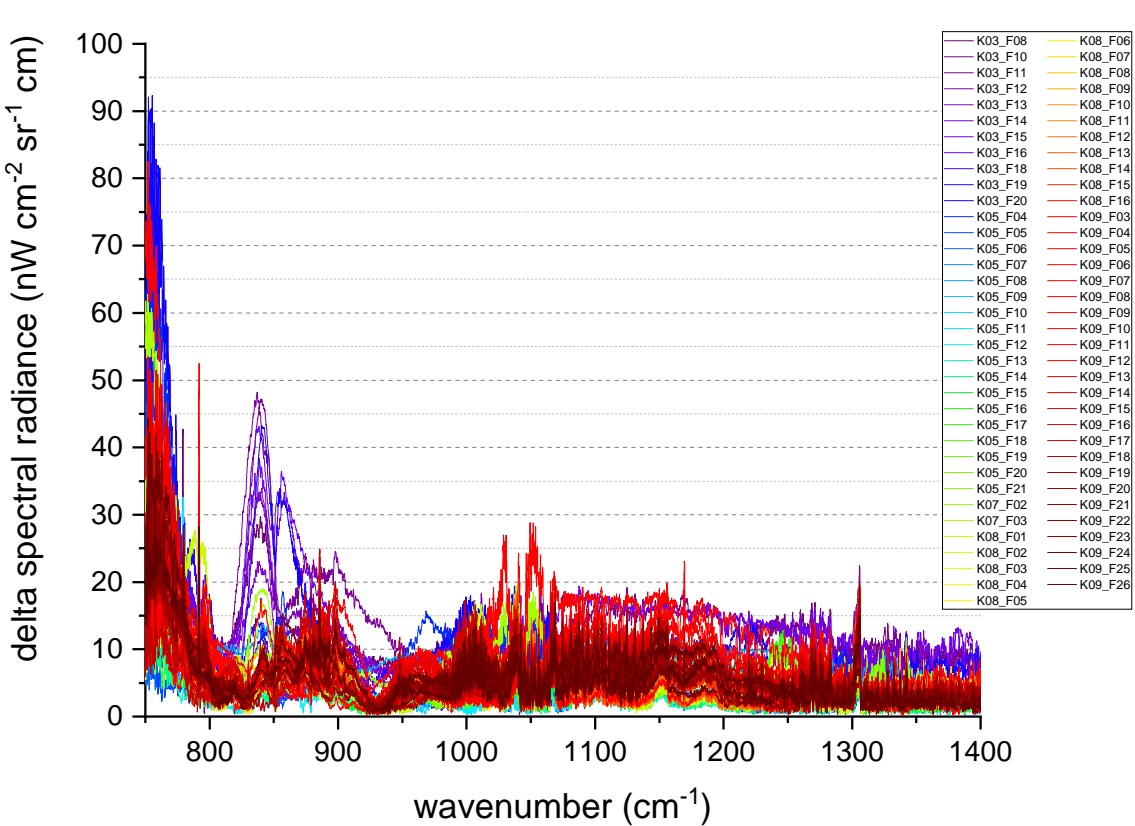

**Figure A4.** 1-sigma standard deviations of the fit residuals for each flight. For details, see text.



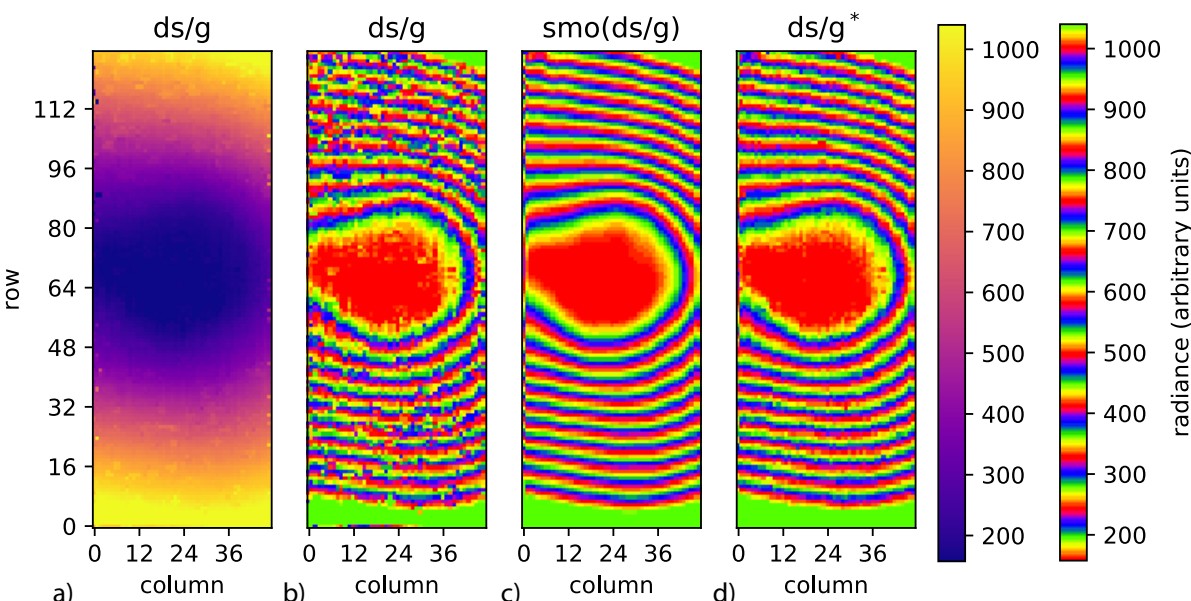

**Figure A5.** All: radiance offset at $964\,\mathrm{cm}^{-1}$. Panel **(a)**: the raw unshaved deep space spectra divided by the uncorrected gain. Panel **(b)**: the same data, but with a color scale emphasizing irregularities. Panel **(c)**: the image smoothened by polynomial fits. Panel **(d)**: the raw unshaved spectra divided by the corrected gain.