# Peer review of "Quantification and mitigation of the airborne limb imaging FTIR GLORIA instrument effects and uncertainties"

_Atmospheric Measurement Techniques, 2021_

## Author Comment (AC1)

We thank the reviewers for their review, interesting questions, and detailed technical correc-2 tions.

All technical suggestions have been applied if not indicated otherwise.

**4 1 Reply to Referee #1**

**5 1.1 Major Comments**

1. My only suggestion would be to explicitly identify any improvement (or not) to the level 2 products if possible.

The actual level 2 data has so far not improved significantly over the last published version of Guggenmoser et al. (2015). There are several improvements w.r.t. to SNR and gain knowledge, which allow for less regularization of level 2 data and different retrieval concepts (e.g. a continuum fit instead of a scale and offset fit). Also some secondary gases with weak emissions should be more accessible; first retrievals for our latest balloon borne campaign show very promising results in this direction.

However, the major advance presented here is the absolute quantification of uncertainties,
 which were previously largely unknown due to our lack of understanding of the non-linearity
 change of the detector.

We reworked the conclusions to better express these points.

**18 1.2 Specific Comments**

1. (a) Can the authors comment on the possible cause of this residual offset?

- (b) It appears that the fit points are close to regions that are devoid of atmospheric signal for the first 4 points (left to right); however, this is less clear with the 5th Can the authors clarify how the micro-windows used in the fit to the Plank function were chosen?
  - (c) A spectral shift is also derived from the shaved deep space measurements. This is calculated separately in each of the micro-windows in Table 2. How large are the individual shifts and how much do they vary between the micro-windows? Is this the same spectral shift that is characterized in Section 5.5?
- (a) We could not nail down this residual offset to a specific physical effect. It may be linked to a non-perfect non-linearity correction in combination with the large extrapolation needed to calculate the instrument offset from the two relatively warm blackbody measurements. Also small errors in the blackbody temperatures may contribute. We have added these possible causes in the text.
  - (b) The micro-windows used for the offset and spectral shift determination were chosen by eye using forward calculated spectra for typical atmospheric situations. Care was taken to avoid broadband atmospheric emission features in these micro-windows in order to clearly identify the (residual) instrumental offset. The selected micro-windows also contain isolated spectral lines to allow for the determination of a spectral shift. The KOPRA fit takes the spectral lines within the micro-windows into account during offset determination. We have added some explanation on the microwindows in the text.
- (c) The shift is usually in the order of 2–3 ppm, with a slow variation over wavenumber. It is the same shift that is characterized in section 5.4, only the determination method is slightly different. In section 5.4 we use the algorithms described by Kleinert et al.
  (2014), while in this section, we use the shift fit algorithm implemented in KOPRAFIT (Höpfner et al., 1998). The results obtained with the different methods are consistent.

2. Line 176: Why not just remove the bad pixels and use the mean of the central row to provide a high SNR spectrum for the central pixel as opposed to the median?

As it is demonstrated in section 4.5, it is difficult to strictly classify good and bad pixels; moreover, a considerable number of pixels changes its behaviour with time. Therefore it is simpler and more robust to use the median over all pixels, and it could be shown that the spectrum obtained by this method is representative for the central pixel row.

8 3.

Line 204: The authors refer the reader to Figure 2 regarding the linear interpolation between rows. However, only the median spectrum is shown in Figure 2. Can the authors clarify what was being referred to here?

The measured median spectrum is shown in black. The red curve shows the forward calculation for the central pixel row, using the fit results as described before. With the same fit results, forward spectra are calculated for the lowermost and the uppermost row (not shown). When interpolating the spectra linearly between the lowermost and the uppermost row, one obtains an interpolated spectrum for the central row. This is shown in blue. This spectrum is very close to the red one, therefore it cannot be distinguished in the upper part of the figure, as indicated in the figure caption. The red and the blue curve are only distinguishable in the residual plot in the lower part of the figure, showing that the linear interpolation between lowermost and uppermost row is a very good approximation to the spectrum that is obtained for the elevation angle of the central row. We have re-phrased the corresponding paragraph in order to make this more clear.

Figure A4: The data plotted in Figure A4 is used to characterize the quality of 4. 21 the removal of the atmospheric signatures from the deep space measurements. The 22 authors note large deviations near 830 cm-1 that were attributed to the germanium 23 window emission that was not corrected until after the early TACTS campaigns. 24 However, from Figure A4, there are also enhanced features between 750 cm-1 to 25 800 cm-1, as well as, near 1050 cm-1 and 1300 cm-1 that don't appear linked to 26 variations in the window emission. Can the authors provide clarity on the cause 27 of these features? 28

The other enhanced features are mainly atmospheric signatures, which are not perfectly removed by our algorithm, e.g., CFC-11 and HNO3 around 830–920 cm-1, O3 near 1050 cm-1, and the CH4 Q-branch at 1304 cm-1. These are, however, to a large extent below the 20 nWcm-2sr-1cm which we estimate as  $2-\sigma$  uncertainty. Below 780 cm-1 the deviations become larger, which is attributed to a strongly increasing NESR, contributions from the window emission and problems with the spectroscopic data. Since our nominal spectral range starts at 780 cm-1, this was not considered in the uncertainty estimation.

We have added a paragraph in the text after line 754, discussing these features.

5. Line 741: What is the reasoning behind using different profiles for the same species in the different micro windows? For example, for ozone, v2 is used for 850 cm-1 to 1065 cm-1 while the other spectral ranges use v0.

This is explained in lines 732ff.: For some species, it was not possible to find a single VMR profile representing the whole spectral range adequately, most likely due to inconsistencies in the spectroscopic data and/or different temperature dependencies.

- Line 319: Why was the beam splitter turned by 90 degrees? During which campaign does Figure 8 correspond? In that case, the parasitic images are still distributed horizontally.
- 46 We added that it was turned due to a manufacturing defect of the beamsplitter.
- A last minute repair of the instrument before the campaign did not leave sufficient time for testing. Thus the defect was only discovered shortly after the back-to-back campaigns.

Figure 8 stems from the WISE campaign with turned beam splitter. The parasitic images of the crescent moon are clearly above (negative image, box B) and below (positive image, not highlighted, vertical pixels 6–12), i.e. vertically distributed.

7.

6

g

Line 365: The wording here suggests that the Gaussian was fit only to the left portion of the distribution in Figure 10. If that is the case, then what criteria was used to reject certain data from the fit?

We indeed used only the portion to the left of the peak for the fit. The distribution is generated by all  $\approx 6000$  pixels of the detector, each of which exhibits a different behaviour, which is mostly Gaussian. The "good" pixels have the lowest deviation, whereas "bad" pixels have — by definition — a larger deviation, i.e. they are shifted towards the right on the plot. This implies that the r.h.s. of the plot is generated by the sum of multiple Gaussian-like distributions and dominated by the bad pixels, while the l.h.s. of the distribution is dominated by the good pixels.

Practically, the distribution is computed in 2000 bins over 0 to 120 nWcm-2sr-1cm. deviation. We smooth it with a sliding window of width 5 and compute the peak position. Then, we discard everything more than 2 nWcm-2sr-1cm. to the right of the peak and use the original data for the fit. The smoothing is necessary to make the algorithm robust against the noise. Adding a small portion of the distribution to the right of the peak seemed to make the algorithm overall more well-behaved, as the difference in outcome is very small with the notable exception of a few flights with unfortunate noise in the vicinity of the peak.

Obviously, this is a qualitative argument and the whole algorithm is an heuristic. But the 22 Gaussian fits the side left of the peak well, which gives empirical evidence to the validity 23 of the method, i.e., we gain some (flawed) insight into the behaviour of the "good" pixels, 24 which can certainly be used to monitor the quality of the detector from flight to flight.

8. Figure 17 (c): There is clear "band" structure in the noise estimate that is most
likely associated with the readout electronics. The variation in the vertical and
impact on retrievals could potentially be minimized by rotating the camera. Was
this considered?

The "band" or stripe pattern is caused by the 8-tap readout circuit of the detector, where each vertical line represents the individual noise characteristics of its corresponding ADC. Changing the camera direction would indeed alleviate this particular problem.

There is, however, "coloured noise" in the detector, which is correlated over 8 vertical pixels (see, e.g., Fig 3b). In the current setup, this is largely mitigated by averaging over 48 uncorrelated columns.

But the most important point, and the original motivation for the current camera orientation, is the detector readout speed. Due to the particulars of the detector electronics, changing the camera orientation would result in a reduction in temporal sampling speed of about 35 "percent", leading to a correspondingly lower interferometer sampling speed and worse velocity stabilization of the interferometer sled.

41 9.

**Line 526: Isn't the correction presented in Section 4.3 supposed to correct for the changing temperature of the window?**

The temperature correction is far from perfect (according to the main text, we believe it to correct  $\approx 90\%$  of the effect). It is, currently, also only applicable to calibrated atmospheric spectra.

In particular, we cannot apply a window correction without having computed a gain. To compute the gain, we average over several consecutive black body and deep space spectra *without* a window temperature correction. We thus discard averaged spectra with large window temperature variations, but there are always small temperature variations, especially during the long deep space measurements. Even if we were able to correct individual uncali-1 brated spectra, there would still remain the non-linear effect of temperature variation within 2 an interferogram acquisition. 3

We have dabbled in even more involved correction schemes, but deemed the added complexity 4 of the algorithm not worth the achieved gains. 5

It is more important that, from this diagnosis, we know the gain to have potentially an 6 increased uncertainty for some flights, which we can leverage to optimise and characterise 7 our level 2 processing schemes. 8

10. Line 623: Was the PSF characterized in the lab or simulated using optical design 9 software? What is the expected PSF and how well does it match the chosen profile? 10

We wanted to state that the derived aperture of the idealized Airy-Disk is a sufficiently good 11 approximation for our instrument design as the optical system is always diffraction limited. 12 We adapted the text.

A direct measurement of the PSF is difficult and has not yet been achieved in a quality 14 and with a confidence allowing the use of the measured parameters. The PSF has neverthe-15 less been indirectly verified through its impact on the instrument spectral response, both in 16 laboratory and in field measurements. These verification confirmed that a simple Airy-disk 17 model is sufficient for a nominally operating instrument. On our todo list is the measure-18 ment of Venus with sufficient (astronomical) pointing stability that may allow us a direct 19 measurement. 20

There are more involved simulations using ZEMAX available, particular to examine the 21 influence of imaging aberrations and abnormal or degraded operation mode (i.e. error on 22 the focal plane adjustment) on spectral line shape (Kretschmer, 2013). These studies showed 23 that the use of a simple physical Airy model is well within the uncertainty of a measured 24 PSF, at least for nominal operation. For degraded operation, as other perturbation dominate 25 the processing errors, the advantage of having a more accurate PSF model is negligible. A 26 different instrument with an optical system not diffraction limited on the whole imaging field 27 or with a wider spectral range may indeed have to use a different PSF model. 28

11. Line 658: I think the vertical structure in these errors could be quite important 29 since the retrieved profiles are vertically resolved. It would be interesting to know 30 which ones have the largest vertical structures. Can you comment on the impact 31 of these structures on the retrieved profiles? 32

The spatial structure of the errors is very important, indeed. We partially mention it in the text, e.g. that the temperature error induced by the elevation uncertainty is negligible at 34 flight altitude, but steadily increases towards lower altitudes. Most errors have some kind of 35 gradient. In addition, elevation uncertainty induces necessarily large errors in the vicinity of 36 small-scale structures, e.g. cirrus clouds due to the shifted position of the derived structure. 37

However, an in depth discussion of level 2 errors can be found in older publications of ours, 38 most recent by Johansson et al. (2018). While the estimated magnitude of some errors has 39 changed (mostly for the better), the qualitative impact has not. 40

**1.3Comments 41**

All suggestions of the reviewer were applied.

**1 2 Reply to Referee #2**

**2 2.1 Comment**

1.

4 5

In the abstract and in other sections of the manuscript, the authors report information on the number of campaigns, total flight track and flight hours of operation of the GLORIA instrument. It would be useful to complement these pieces of information with the total number of flights performed during the eight campaigns.

We added that GLORIA was used on 75 scientific flights so far.

**8 2.2 Technical Corrections**

We addressed all technical corrections of the reviewer with one exception: We kept "an NESR" 10 in lines 495 and 497, because the abbreviation starts with a vowel *sound*.

**11** References**

Guggenmoser, T., Blank, J., Kleinert, A., Latzko, T., Ungermann, J., Friedl-Vallon, F., Höpfner,
M., Kaufmann, M., Kretschmer, E., Maucher, G., Neubert, T., Oelhaf, H., Preusse, P., Riese,
M., Rongen, H., Sha, M. K., Sumińska-Ebersoldt, O., and Tan, V.: New calibration noise
suppression techniques for the GLORIA limb imager, Atmos. Meas. Tech., 8, 3147–3161,
https://doi.org/10.5194/amt-8-3147-2015, 2015.

Höpfner, M., Stiller, G. P., Kuntz, M., von Clarmann, T., Echle, G., Funke, B., Glatthor, N., Hase,
F., Kemnitzer, H., and Zorn, S.: Karlsruhe optimized and precise radiative transfer algorithm:
II. Interface to retrieval applications, in: Optical Remote Sensing of the Atmosphere and Clouds,
edited by Wang, J., Wu, B., Ogawa, T., and hua Guan, Z., vol. 3501, pp. 186 – 195, International

Society for Optics and Photonics, SPIE, https://doi.org/10.1117/12.317753, 1998.

Johansson, S., Woiwode, W., Höpfner, M., Friedl-Vallon, F., Kleinert, A., Kretschmer, E., Latzko,
T., Orphal, J., Preusse, P., Ungermann, J., Santee, M. L., Jurkat-Witschas, T., Marsing, A.,
Voigt, C., Giez, A., Krämer, M., Rolf, C., Zahn, A., Engel, A., Sinnhuber, B.-M., and Oelhaf,
H.: Airborne limb-imaging measurements of temperature, HNO3, O3, ClONO2, H2O and CFC12 during the Arctic winter 2015/2016: characterization, in situ validation and comparison
to Aura/MLS, Atmos. Meas. Tech., 11, 4737–4756, https://doi.org/10.5194/amt-11-4737-2018,
2018.

Kleinert, A., Friedl-Vallon, F., Guggenmoser, T., Höpfner, M., Neubert, T., Ribalda, R., Sha,
M. K., Ungermann, J., Blank, J., Ebersoldt, A., Kretschmer, E., Latzko, T., Oelhaf, H.,
Olschewski, F., and Preusse, P.: Level 0 to 1 processing of the imaging Fourier transform
spectrometer GLORIA: generation of radiometrically and spectrally calibrated spectra, Atmos.
Meas. Tech., 7, 4167–4184, https://doi.org/10.5194/amt-7-4167-2014, 2014.

Kretschmer, E.: Modelling of the Instrument Spectral Response of Conventional and Imaging
Fourier Transform Spectrometers, Dissertation, Université Laval, 2013.